# Diff-PIC: Revolutionizing Particle-In-Cell Nuclear Fusion Simulation with Diffusion Models

**Chuan Liu[1], Chunshu Wu[1], Shihui Cao[1], Mingkai Chen[2], James Chenhao Liang[2],**
**Ang Li[3], Michael Huang[1], Chuang Ren[1], Ying Nian Wu[4], Dongfang Liu[2], Tony (Tong) Geng[1]**
[1]University of Rochester, [2]Rochester Institute of Technology,
[3]Pacific Northwest National Laboratory, [4]University of California, Los Angeles

## Abstract

The rapid development of AI highlights the pressing need for sustainable energy, a critical global challenge for decades. Nuclear fusion, generally seen as a promising solution, has been the focus of intensive research for nearly a century, with investments reaching hundreds of billions of dollars. Recent advancements in Inertial Confinement Fusion (ICF) have drawn significant attention to fusion research, in which Laser-Plasma Interaction (LPI) is critical for ensuring fusion stability and efficiency. However, the complexity of LPI makes analytical approaches impractical, leaving researchers dependent on extremely computationally intensive Particle-in-Cell (PIC) simulations to generate data, posing a significant bottleneck to the advancement of fusion research. In response, this work introduces Diff-PIC, a novel framework that leverages conditional diffusion models as a computationally efficient alternative to PIC simulations for generating high-fidelity scientific LPI data. In this work, physical patterns captured by PIC simulations are distilled into diffusion models associated with two tailored enhancements: (1) To effectively capture the complex relationships between physical parameters and their corresponding outcomes, the parameters are encoded in a physically informed manner. (2) To further enhance efficiency while maintaining physical validity, the rectified flow technique is employed to transform our model into a one-step conditional diffusion model. Experimental results show that Diff-PIC achieves a $\sim16{,}200\times$ speedup compared to traditional PIC on a 100 picosecond simulation, while delivering superior accuracy compared to other data generation approaches.

## 1 Introduction

Sustainable energy stands as one of the paramount challenges of our era, particularly with the rapid advancement of AI. The recent successful demonstration of fusion ignition (Abu-Shawareb et al., 2024) underscores the potential of fusion as a sustainable energy source. In 2023 and 2024, the National Ignition Facility (NIF) achieved groundbreaking milestones, generating 3.4 MJ and 5.2 MJ of fusion energy from 2.2 MJ input energy, respectively. Given that the estimated output could reach $\sim$120 MJ (Suter et al., 2004), there is a growing demand for a deeper understanding of the fundamental science behind ignition, especially the physics governing the interaction between the laser and the plasma emitted when the laser bombards the fuel pellet. However, LPI is a complex problem that is traditionally analyzed using time-stepping method, particularly through PIC simulations (Tskhakaya et al., 2007; Langdon, 2014; Arber et al., 2015). Despite being the preeminent standard for modeling the physics of LPI, PIC simulations are exceedingly intensive in computation, often requiring tens of millions of CPU hours to obtain meaningful outputs. The computational overhead of PIC simulations has become a daunting bottleneck in fusion research, raising the pressing need for innovative methodologies capable of generating high-quality scientific data with substantially reduced computational burden.

Over the years, numerous CPU-GPU based accelerations have been developed for PIC simulations (Bowers et al., 2008; Sgattoni et al., 2015). Although invaluable, these efforts remain within the scope of the time-stepping approach that iteratively executes over infinitesimal time intervals, falling short in addressing the inherent computational overhead of long-term simulations. Recent

advancements in generative AI – diffusion models, however, present a novel approach to bypass the constraint. Diffusion models have demonstrated exceptional capabilities in Computer Vision (CV), synthesizing highly complex data distributions that match real data with high fidelity (Ho et al., 2020; Song & Ermon, 2019). Rooted in the diffusion concept in physics, diffusion models can be understood as constructing a highly complex field that governs the evolution of variables, analogous to the motion of particles in cells. This has sparked significant interest in their potential for generating scientific data, as recent applications of diffusion models in molecular dynamics simulations demonstrated (Wu & Li, 2023; Petersen et al., 2023).

Although diffusion models exhibit promising potential for generating PIC simulation data, two critical research gaps remain. ❶ *Physical soundness:* It remains unclear how to effectively integrate continuous physical parameters into diffusion models to generate data that faithfully represent the underlying physics. ❷ *Substantial efficiency improvement:* Although the requirement for infinitesimal time intervals in PIC simulations has been relaxed, the step-by-step denoising process in diffusion models is still computationally demanding, limiting the practicality of diffusion models as advanced alternatives to PIC simulations.

To address these challenges, we propose **Diff-PIC**, which leverages diffusion models to efficiently generate a snapshot of arbitrary time, under arbitrary physical parameters within certain ranges. Specifically, ❶ we develop a conditional diffusion model with a Physically-Informed Parameter Encoder. This encoder enables the model to capture the relationship between continuous physical parameters and PIC simulation data, effectively distilling the underlying physics into Diff-PIC. ❷ We employ the rectified flow technique (Liu et al., 2022) to eliminate the requirement for multistep denoising, further optimizing the runtime efficiency of Diff-PIC. As highlighted in Fig. 1, Diff-PIC achieves orders-of-magnitude speedup com-

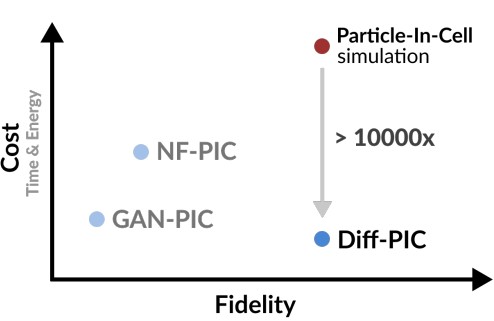

Figure 1: Overview of this work.

pared to PIC simulations, and delivering superior fidelity compared to baselines "GAN-PIC" and "NF-PIC". In summary, our work represents the first known effort to tackle the imperative challenges associated with generating high-quality PIC simulation data for LPI using diffusion models. The core contributions of our work include:

- We propose Diff-PIC, a pioneering study that utilizes diffusion models as a computationally efficient alternative to PIC simulations, accelerating the scientific data generation for nuclear fusion research.
- We develop a physically-informed conditional diffusion model that seamlessly integrates physical parameters into the diffusion model. The designed condition encoder effectively facilitates the generalization of simulation data within and beyond seen parameters, endowing the model with robust generalization capabilities.
- We implement the rectified flow technique to make our model a one-step diffusion model, further enhancing its efficiency in generating high-fidelity fusion data.
- Experimental results demonstrate that our method achieves a remarkable speedup of ∼16,200 times compared to traditional PIC simulations while preserving physical validity of the generated data.

## 2 BACKGROUND

**Inertial Confinement Fusion** (ICF) is a method of achieving controlled nuclear fusion by using intense energy pulses to heat small fuel pellets (Keefe, 1982; Betti & Hurricane, 2016), typically containing isotopes of hydrogen such as deuterium and tritium. This process unfolds the nuclear fusion reaction as delineated below:

$$^{2}\text{H} + {}^{3}\text{H} \rightarrow {}^{4}\text{He} + {}^{1}\text{neutron} + \textbf{Energy}. \tag{1}$$

Given the ubiquity of these hydrogen isotopes, nuclear fusion holds immense potential to provide "near-infinite" energy by achieving the necessary temperature and pressure conditions to initiate fusion

reactions, attaining a positive energy gain (*i.e.*, output energy surpassing the input). To optimize the sophisticated initiation of fusion, an advanced understanding of the underlying LPI is essential. For this purpose, PIC simulation is considered a crucial tool for providing theoretical insights into LPI, due to its capability to predict and interpret physical phenomena.

**PIC Simulations** are widely used in the study of plasma physics and fusion energy research (Tskhakaya et al., 2007; Derouillat et al., 2018). Developed in the mid-$20th$ century, the PIC method has become a cornerstone in the simulation of complex plasma behaviors, enabling researchers to delve into the underlying physics. To highlight, PIC is especially useful in LPI studies (Arber et al., 2015; Klimo et al., 2010), which involve complex dynamics of electrons and ions. PIC simulations track the trajectories and interactions of these charged particles under the influence of electromagnetic fields, providing insights for shock wave formation and heating mechanisms that are essential for ICF. In essence, PIC is an iterative time-stepping method applied to atomic particles such as electrons and ions. Within each iteration, particles are systematically arranged into discrete cells according to their spatial distribution, with their positions and velocities being updated over infinitesimally small time steps. To ensure accuracy, simulating LPI over only a few hundred picoseconds (*i.e.*, $10^{-10}$ seconds) requires hundreds of thousands of sophisticated PIC iterations. This imposes significant demands on computational storage and processing capabilities. As a result, the PIC methodology has emerged as a stringent bottleneck in fusion research, significantly restricting progress in this domain.

**Diffusion models** have emerged as powerful generative models, capable of generating high-fidelity data (Ho et al., 2020; Song & Ermon, 2019). Diffusion models involve two processes: the forward process and the reverse process. In the forward process, Gaussian noise is gradually added to clean data, transforming the original data distribution into a tractable Gaussian distribution. Mathematically, this can be represented as:

$$q(\mathbf{x}_t|\mathbf{x}_{t-1}) = \mathcal{N}(\mathbf{x}_t; \sqrt{\alpha_t}\mathbf{x}_{t-1}, (1 - \alpha_t)\mathbf{I}), \tag{2}$$

where $\alpha_t$ regulates the noise level at each time step $t$. The reverse process reconstructs the original data from the noisy distribution by iteratively denoising the data. This procedure is typically parameterized by a neural network, which is trained to approximate the inverse of the forward diffusion process:

$$p_\theta(\mathbf{x}_{t-1}|\mathbf{x}_t) = \mathcal{N}(\mathbf{x}_{t-1}; \mu_\theta(\mathbf{x}_t, t), \Sigma_\theta(\mathbf{x}_t, t)), \tag{3}$$

where $\mu_\theta$ represent the mean predicted by the neural network and $\Sigma_\theta$ denotes time-dependent parameters that control the variance or uncertainty at each denoising step.

## 3 DIFF-PIC

This section introduces Diff-PIC, a physically-informed conditional diffusion model tailored for generating high-fidelity synthetic data for LPI in nuclear fusion. As illustrated in Fig. 2, accepting physical parameters as inputs, the parameters are encoded and integrated with the model through the physically informed parameter encoder. Additionally, the rectified flow method is adopted to break free from the multi-step denoising process in diffusion models, unlocking the model's capability to generate scientific data in one single step.

### 3.1 THE OVERALL DISTILLATION FRAMEWORK

Since LPI is governed by the behavior of electromagnetic fields that dominate plasma dynamics, as a representative case, the task assigned to Diff-PIC is to generate high-fidelity 2D electric fields under various physical parameters, which is also a standard practice in PIC simulations. Specifically, we focus on the following critical physical parameters in LPI: "Electron Temperature ($T_e$) in $keV$," "Ion Temperature ($T_i$) in $keV$," and "Laser Intensity ($I$) in $W/m^2$." Additionally, to enable direct generation of the electric field at a specific simulation time (a snapshot), we include an additional parameter $t_{as}$ that represents the simulation time for an arbitrary snapshot. Through training, the model captures the relationships between the four parameters and the resulting electric fields. Once trained, the model takes the four parameters as inputs and produces electric field snapshots $\mathbf{E}(t_{as}, \theta)$ corresponding to the specified physical parameters $\theta = \{T_e, T_i, I\}$. These generated snapshots can be used for multiple purposes, including data augmentation and parameter exploration. Essentially, the proposed distillation framework offers two advantages:

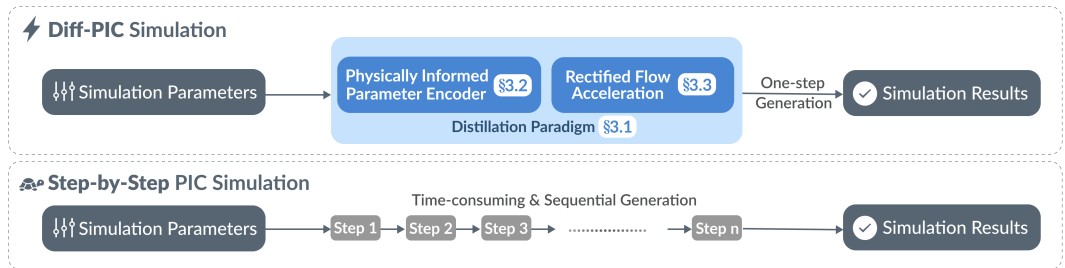

Figure 2: Workflow of the proposed Diff-PIC compared to traditional step-by-step PIC simulations.

- *Systemic efficiency.* Unlike traditional PIC simulations that generate data sequentially over time, the proposed diffusion model can directly produce data for any target snapshots by varying $t_{as}$. This non-sequential behavior allows for substantially more efficient data generation and analysis (see Table 3), enabling researchers to focus on specific times of interest without needing to simulate the entire LPI process from the beginning.

- *Data dependency relaxation.* We treat snapshots from various parameter combinations as distinct distributions. This approach decouples data dependency in model training, enabling the model to efficiently learn from individual snapshots while generalizing across a wide array of parameters.

## 3.2 PHYSICALLY-INFORMED PARAMETER ENCODER

PIC simulations employ continuous physical parameters as inputs, which necessitates a seamless and continuous transition in the resulting synthesized data as input parameters are adjusted. Consequently, an encoder is considered exceptionally important in this scenario, responsible for transforming domain-specific inputs into effective embeddings. In particular, these inputs comprise the simulation parameters $\theta$ and the target simulation snapshot $t_{as}$. Furthermore, the encoder should also excel in both *interpolation* and *extrapolation* — critical measures of the model's ability of generalization. Interpolation capability refers to the encoder's proficiency in generating suitable embeddings for new parameters that, although not encountered during training, lie within the range of parameters seen during training. Extrapolation capability, conversely, pertains to generating embeddings for parameters that fall outside the range of those seen during training. Notably, both capabilities are indispensable for the LPI problem, in order to explore a large enough parameter space to provide sufficient insight into further LPI evaluations.

To meet the essentials, we introduce a Physically-Informed Parameter Encoder (PIPE) as shown in Fig. 3. To encode the simulation parameters $\theta$, we employ two distinct types of encoders tailored for interpolation and extrapolation tasks. For interpolation, we employ Positional Encoding (Vaswani et al., 2017) (denoted by "∼" in a circle), which leverages sinusoidal functions to encode the input parameters, facilitating smooth transitions between observed parameters. To enhance extrapolation capability, we incorporate a polynomial encoder ("P" in a circle). Polynomial encoders are widely used to approximate a wide range of functions effectively, capturing nonlinear correlations and unbounded growth patterns in the data, which is crucial for extrapolation. This is achieved through transformation constructed as a linear combination of polynomial basis functions $f_i(\theta)$ of varying degrees:

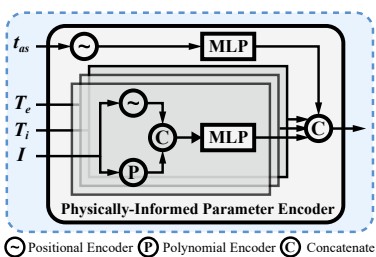

Figure 3: The proposed PIPE.

$$\mathcal{P}(\theta) = \sum_{i=1}^{n} f_i(\theta), \qquad (4)$$

where $n$ denotes the maximum order of polynomial terms, and the polynomial $\mathcal{P}(\theta)$ can be chosen as Chebyshev polynomials and Legendre polynomials, based on the characteristics of the parameter space. This polynomial enhancement allows the encoder to generate plausible embeddings for parameters beyond those encountered during training, ensuring robust performance across a broader spectrum of simulation scenarios. Subsequently, we concatenate ("C" in a circle) the embeddings from

these two encoders and apply a Multi-Layer Perceptron (MLP) to further refine the embeddings. The MLP, with its trainable parameters, learns to combine and transform the concatenated embeddings, resulting in a more informative representation of the input parameters. For encoding the simulation time step $t_{as}$, we utilize Positional Encoding (Vaswani et al., 2017) followed by an MLP layer. This approach is specifically chosen to learn continuous representations that facilitate smooth transitions between consecutive snapshots, thereby enhancing the model's temporal coherence. In summary, this design offers the following advantages:

- *Algorithmic generalization.* PIPE improves the generalizability of the conditional diffusion model (see Table 1 and Table 2). The dual-encoding strategy captures nonlinear relationships by incorporating both positional and polynomial encoders, empowering the model to adeptly manage a diverse array of parameters, ranging from scenarios encountered during the training phase to parameters that lie beyond the spectrum of the training data.

- *Adaptive transferability.* By fine-tuning the pre-trained encoder with a new dataset, it facilitates adaptation to other related tasks.

### 3.3 RECTIFIED FLOW ACCELERATION

To enable rapid generation of high-fidelity synthetic data, we employ the Rectified Flow Acceleration (RFA) technique. Based on the principles of rectified flow (Liu et al., 2022; 2024), RFA converts the original complex denoising trajectory from initial noise $\epsilon$ to the target electric field snapshot $\mathbf{E}(t_{as}, \theta)$ into a direct and straight path. During training, Diff-PIC minimizes the following objective function:

$$\arg\min_{\zeta} \quad \mathbb{E}\left[\int_0^1 \|(\mathbf{E}(t_{as}, \theta) - \epsilon) - \zeta(\mathbf{E}_t, t \mid t_{as}, \theta)\|^2 \, dt\right], \tag{5}$$

where $\mathbf{E}_t = t\mathbf{E}(t_{as}, \theta) + (1 - t)\epsilon$ denotes the linear interpolation between $\mathbf{E}(t_{as}, \theta)$ and $\epsilon$ across the diffusion timeline, with $t$ ranging from 0 to 1. Using a modified U-Net backbone, $\zeta$ learns the denoising trajectory by minimizing the expectation of the squared deviations between $\mathbf{E}(t_{as}, \theta) - \epsilon$ and $\zeta(\mathbf{E}_t, t \mid t_{as}, \theta)$. Once $\zeta$ is trained, we further straighten the learned trajectories through an interactive reflow procedure (Liu et al., 2024). In summary, the RFA module provides additional benefits for our method:

- *Streamlined denoising process.* RFA significantly accelerates the denoising process (see Table 3) by converting the complex multi-step denoising process into a one-step denoising process.

- *Robust optimization.* By streamlining the denoising process, RFA reduces the possibility of error accumulation that occurs with the multi-step denoising trajectory.

## 4 EVALUATION

### 4.1 EXPERIMENTAL SETUP

**Datasets.** We construct a dataset comprising 6,615 simulations across varied physical parameters, each containing 80 snapshots of electric fields along two orthogonal directions denoted as E1 and E2. The data are generated using OSIRIS (Fonseca et al., 2002), a well-established PIC simulation software suite. The dataset encompasses diverse physical parameters, including $T_e$, $T_i$, and $I$, all of which are critical in influencing the resultant electric fields.

**Metrics.** To validate physical soundness, Mean Absolute Error (MAE) and Root Mean Squared Error (RMSE) are used to evaluate the electric field difference and the energy difference between the Diff-PIC generated data and the ground truth produced by PIC simulations. To further evaluate the difference in the generated and the ground truth data distributions, the Fréchet Inception Distance (FID) metric is also employed.

**Baselines.** We compare Diff-PIC with two other types of generative models (Karras et al., 2020; Zhang & Chen, 2021). Since neither of them originally supports learning meaningful embeddings for the physical parameters, for fair comparison, we equip them with the proposed PIPE to establish two baselines: GAN-PIC and NF-PIC. More details are provided in Appendix A.2.

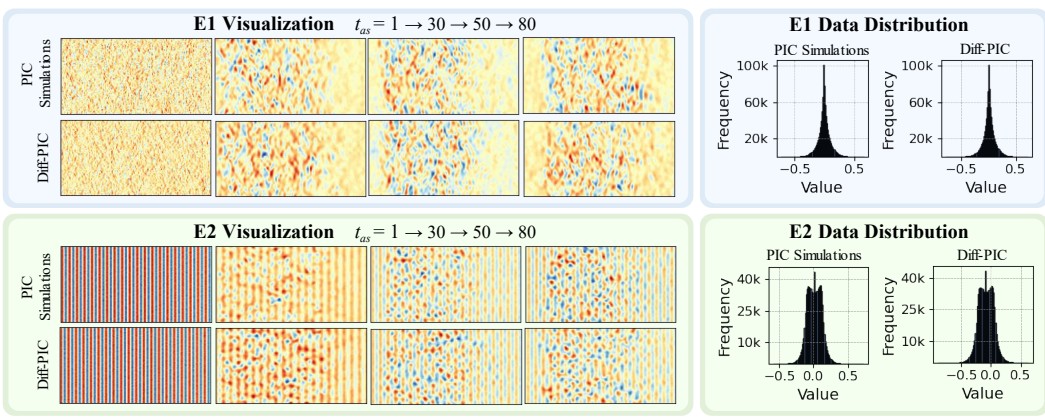

Figure 4: **Visualization and Comparison** of PIC simulations and Diff-PIC.

Table 1: **Quantitative results** for interpolation evaluation. MAE and RMSE are on the order of $10^{-2}$.

| Method | Training Set for E1 | | | Testing Set for E1 | | | Training Set for E2 | | | Testing Set for E2 | | |
|---|---|---|---|---|---|---|---|---|---|---|---|---|
| | MAE↓ | RMSE↓ | FID↓ | MAE↓ | RMSE↓ | FID↓ | MAE↓ | RMSE↓ | FID↓ | MAE↓ | RMSE↓ | FID↓ |
| **GAN-PIC** | 4.59 | 5.31 | 2.32 | 4.73 | 5.84 | 2.51 | 1.82 | 2.07 | 0.973 | 1.97 | 2.18 | 1.03 |
| **NF-PIC** | 4.46 | 5.12 | 2.06 | 4.61 | 5.35 | 2.42 | 1.70 | 2.03 | 0.914 | 1.86 | 2.45 | 0.986 |
| **Diff-PIC** | 1.56 | 2.67 | 1.21 | 1.68 | 2.29 | 1.62 | 0.795 | 0.932 | 0.328 | 0.826 | 1.03 | 0.341 |

**Diff-PIC Configurations.** The architectural foundation of our model is a modified U-Net framework (Ronneberger et al., 2015), comprising three down-sampling blocks and three up-sampling blocks, strategically integrated with attention mechanisms to capture both local and global dependencies. For parameter encoding, the positional encoders generate 16-dimensional embeddings using sinusoidal functions, facilitating smooth interpolation of input parameters. Polynomial encoders incorporate polynomial terms up to the fourth degree to capture nonlinear relationships, resulting in a comprehensive 20-dimensional embedding for each parameter in $\theta$ after concatenation and transformation via a single-layer MLP. These refined embeddings are concatenated with the simulation data and fed into the U-Net backbone. More implementation details are provided in Appendix A.1.

## 4.2 MAIN RESULTS

In this section, we evaluate Diff-PIC on three key aspects for comprehensive comparisons. (1) The interpolation ability and extrapolation ability. (2) The physical validity of the Diff-PIC generated data. (3) The speedup and energy efficiency compared to traditional PIC simulations.

**Interpolation and Extrapolation.** To evaluate the interpolation capability of Diff-PIC, we sample a specified range for each simulation parameter ($T_e$, $T_i$, and $I$), totaling 500 simulations and $500 \times 80 = 40,000$ snapshots. Then, we randomly split these 500 simulations into training and testing set with the ratio of 80% and 20%. We train Diff-PIC and baselines on the training set, and report the performance of Diff-PIC on the training and testing set in Table 1. To better illustrate the relative error, the metrics are evaluated after normalizing the generated data to the range of [0,1]. The reasonably low MAE, RMSE, and FID scores indicate that the proposed Diff-PIC is able to synthesize high-quality scientific data similar to what PIC generates, while significantly outperforming baselines on all three metrics. On average, Diff-PIC achieves a 59.25% reduction in MAE, a 57.77% reduction in RMSE, and a 49.21% reduction in FID across all testing sets compared to the baselines.

In addition, Fig. 4 compares the results of one randomly selected simulation produced by Diff-PIC and PIC simulations in the testing set, respectively. Throughout the snapshots (e.g., $t_{as} = 1 \rightarrow 30 \rightarrow 50 \rightarrow 80$), the visualizations of the synthetic data closely follow the ground truths, indicating that the physical evolution is preserved over time. Additionally, the distributions on the right demonstrate that our proposed model successfully captures the data distributions of the ground truth.

Table 2: **Quantitative results** for extrapolation. MAE and RMSE are on the order of $10^{-2}$.

| Method | E1 10% | | | E1 20% | | | E2 10% | | | E2 20% | | |
|---|---|---|---|---|---|---|---|---|---|---|---|---|
| | MAE↓ | RMSE↓ | FID↓ | MAE↓ | RMSE↓ | FID↓ | MAE↓ | RMSE↓ | FID↓ | MAE↓ | RMSE↓ | FID↓ |
| **GAN-PIC** | 5.24 | 6.32 | 1.97 | 5.73 | 6.98 | 2.15 | 2.18 | 3.25 | 1.15 | 2.93 | 3.89 | 1.29 |
| **NF-PIC** | 4.74 | 5.31 | 1.85 | 5.41 | 6.46 | 2.08 | 1.90 | 3.16 | 1.04 | 2.63 | 3.42 | 1.17 |
| **Diff-PIC** | 1.83 | 2.40 | 1.74 | 2.18 | 2.62 | 1.82 | 0.947 | 1.36 | 0.536 | 1.13 | 1.85 | 0.673 |

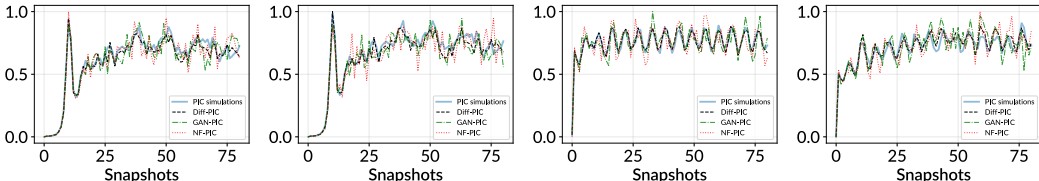

Figure 5: **Energy evaluation** of electric fields for training and test sets with 80 snapshots. From left to right: E1 Training Set, E1 Testing Set, E2 Training Set, and E2 Testing Set.

In terms of extrapolation capability evaluation, we progressively extend the range of physical parameters selected for training. In particular, the ranges are extended by 10% and 20% respectively as case studies. The results in Table 2 demonstrate that Diff-PIC achieves an approximately 2% relative absolute error in extrapolation tasks, significantly outperforming other generative counterparts. On average, Diff-PIC achieves a 59.16% reduction in MAE, a 56.53% reduction in RMSE, and a 29.70% reduction in FID across all test cases compared to the baselines.

**Physical Validity.** In addition to ensuring the high quality of the synthetic data, we assess the physical validity of the generated electric fields by evaluating their energy evolution—a fundamental property that characterizes physical systems. In particular, we randomly select simulations from both the training and testing sets for E1 and E2. The energy profiles of these electric fields are depicted in Fig. 5, where we compare the ground-truth PIC simulations with synthetic data generated by Diff-PIC, GAN-PIC, and NF-PIC. Notably, Diff-PIC closely aligns with the ground truth, demonstrating low errors across all datasets. Specifically, for the E2 electric field, Diff-PIC accurately preserves the energy oscillations in the ground truth, outperforming baselines. This evaluation highlights that Diff-PIC is capable of generating sequentially continuous data, demonstrating the effectiveness of the proposed distillation paradigm and the Physically-Informed Parameter Encoder.

**Speedup and Energy Consumption Reduction.** In addition to traditional PIC simulations, baselines are also included in the comparison shown in Table 3. The PIC simulations are executed on the Perlmutter supercomputer at the National Energy Research Scientific Computing (NERSC) facility, utilizing AMD EPYC 7763 CPUs. Since essential fusion phenomena typically appear at approximately 100 ps, the cost of PIC simulation at 100 ps is selected as the reference for comparisons. The GPU results for Diff-PIC and the baselines are obtained on an Nvidia RTX 4090 GPU to demonstrate the accessibility of this approach to general users. The CPU results for these approaches are acquired on an Intel 13th Gen i9-13900KF CPU. The results highlight that Diff-PIC-GPU achieves a speedup of $\sim 16,200\times$ compared to traditional PIC simulations, along with a similar orders-of-magnitude reduction in energy consumption.

**Ablation Studies on PIPE.** To further demonstrate the effectiveness of PIPE, we conduct ablation studies by replacing the PIPE component in Diff-PIC with other commonly used encoders: MLP and Transformer decoder. Detailed implementations and comparison results are presented in Appendix A.3. The results show that PIPE outperforms both MLP and Transformer decoder in both

Table 3: Speedup comparison across different methods.

| Method | PIC-100 ps | **Diff-PIC**-GPU | **Diff-PIC**-CPU | GAN-PIC-GPU | GAN-PIC-CPU | NF-PIC-GPU | NF-PIC-CPU |
|---|---|---|---|---|---|---|---|
| **Speedup** | 1.00× | 1.62e4× | 519× | 1.56e4× | 523× | 9.21e2× | 24× |
| **Energy Reduction** | 1.00× | 1.01e4× | 1.05e3× | 1.13e4× | 1.47e3× | 8.14e2× | 53× |

interpolation and extrapolation tasks. MLP struggles with both interpolation and extrapolation. The Transformer decoder, while powerful for sequence modeling and capturing relationships in discrete token sequences, is not inherently designed to effectively process continuous physical parameters, resulting in inferior performance.

In addition, to provide more insight into the effectiveness of PIPE, we conduct another ablation study to evaluate the contributions of the positional encoder and polynomial encoder separately. Detailed implementations and comparison results are shown in Appendix A.3, highlighting the individual importance of these encoders. Specifically, combining both positional and polynomial encoders consistently outperforms using either encoder alone in both interpolation and extrapolation tasks. In interpolation tasks, the positional encoder demonstrates better performance than the polynomial encoder, suggesting its important role in interpolation scenarios. Conversely, in extrapolation tasks, particularly at larger ranges (20%), the polynomial encoder shows relatively better performance, indicating its importance for extrapolation capabilities. These results validate our design choice of combining both encoders in PIPE to leverage their complementary strengths.

## 4.3 DISCUSSION

To highlight the value of this work, it is worth noting that our approach exhibits outstanding scalability compared to traditional PIC simulations. In PIC, for $N$ particles, the computing complexity can generally reach a formidable $TN\log N$ with $T$ time steps. In contrast, Diff-PIC is not sensitive to the number of particles in space or the number of time steps. For larger particle spaces, the speedup achieved by Diff-PIC can be easily increased by extra orders of magnitude, further accelerating the research of fusion, or other research areas involving large-scale PIC simulations.

Furthermore, to provide insight into the quality of the synthetic data generated by Diff-PIC and its applicability for ICF research, we reference a SOTA ICF modeling method (Ejaz et al., 2024), which suggests a prediction error of approximately 12% is considered effective. In our experiments, Diff-PIC achieves MAE and RMSE values of approximately 1-2% relative to the PIC simulations. This high level of precision is sufficient for the generated data to be useful in ICF research, such as preliminary analysis, parameter exploration, and prediction modeling.

Additionally, Diff-PIC shows new insights for machine learning methods in science by using diffusion models to address common challenges in complex scientific simulations. Our work demonstrates the power and effectiveness of Diff-PIC as a computationally efficient alternative to expensive ICF simulations. This achievement not only validates the effectiveness of diffusion models in enhancing simulation efficiency but also paves the way for diffusion models to augment or potentially replace a wide range of computationally expensive scientific simulations, enabling more efficient and scalable simulation methods.

## 5 RELATED WORK

**Particle-in-Cell Simulations** have long been fundamental to modeling physical processes in fusion research (Taccogna & Minelli, 2017; Garrigues et al., 2016). However, the computational intensity of PIC simulations presents significant challenges (Verboncoeur, 2005). To mitigate these computational constraints, various methods have explored GPU and hybrid CPU-GPU acceleration technologies. Studies such as (Abreu et al., 2010; Burau et al., 2010; Decyk & Singh, 2011; Kong et al., 2011; Suzuki et al., 2011) have utilized parallel computing, high memory bandwidth, and multiple processors to accelerate simulations. Architecturally, the simulator optimized for the Kepler GPU architecture, as discussed in (Shah et al., 2017), underscores the potential of specific GPU architectures to enhance simulation efficiency. For more intricate simulations, research efforts like (Xu et al., 2012; Chen et al., 2012) have developed hybrid CPU-GPU implementations, and (Wang & Zhu, 2021) introduced a hybrid approach for multi-core and multi-GPU systems, highlighting the continuous integration and evolution of these technologies in accelerating PIC simulations. Despite these advancements, these approaches remain reliant on the fundamental PIC framework, which may not completely address the computational burden due to the inherent time-stepping property of the PIC method. In recent years, rapid advancements in deep learning have opened new pathways for accelerating scientific simulations. Machine learning-based approaches have emerged, such as predicting a vector space that approximates the PIC solutions (Kube et al., 2021), and learning the probability of interactions

between potential collision pairs (Bilbao et al., 2022). However, the approach by (Kube et al., 2021) depends on a pre-computed vector space and may not generalize well to novel scenarios, while (Bilbao et al., 2022)'s method focuses on binary interactions, overlooking the complex many-body interactions in PIC simulations.

Contrastingly, our proposed method overcomes these limitations by employing a conditional diffusion model to distill the complex physics captured by PIC simulations. By leveraging diffusion models, our approach can efficiently generate high-fidelity synthetic data (see Fig. 4 and Table 1) without the computational expense of traditional PIC algorithms (see Table 3). This results in a significant reduction in computational cost while maintaining high simulation accuracy. Moreover, our method is highly adaptable, as it can be fine-tuned with minimal additional data to suit various physical parameters. This flexibility not only renders our approach suitable for a wide range of applications in fusion research but also extends its potential use to other related domains where efficient and accurate simulations are crucial for understanding complex physics.

**Diffusion Models in Scientific Research** have emerged as a promising tool across a myriad of scientific domains. In materials science and chemistry, diffusion models have been applied to enhance molecular dynamics simulations. For example, (Wu & Li, 2023) introduced a diffusion model for molecular dynamics simulations, showing good results in generating molecular trajectories. (Arts et al., 2023) presented an approach that integrates diffusion models with coarse-grained molecular dynamics to develop a new force field to simulate protein dynamics. (Duan et al., 2023) introduced an object-aware diffusion model capable of rapidly generating accurate 3D transition state structures, significantly reducing the computational burden typically associated with quantum chemistry calculations. In astrophysics, diffusion models have been employed to generate realistic galaxy images. (Smith et al., 2022) demonstrated that diffusion models could produce highly detailed and accurate representations of galaxies, which are essential for the analysis of large-scale sky surveys and for enhancing the understanding of the universe. Diffusion models have also found applications in climate science and earth system modeling. For instance, (Oyama et al., 2023) employed a deep generative model to enhance the spatial resolution of global climate simulations, which is crucial for long-term climate projections and infrastructure development planning. (Li et al., 2024) explored the generative emulation of weather forecast ensembles with diffusion models, illustrating their effectiveness as scalable and cost-efficient alternatives to traditional ensemble forecasts, thus improving the reliability and accuracy of predictions for extreme weather events. In particle physics, (Imani et al., 2024) introduced a diffusion model for generating high-quality Liquid Argon Time Projection Chamber (LArTPC) images, showcasing the model's ability to handle the challenges of sparse but locally dense particles.

While these studies highlight the expanding interest and application of diffusion models across various scientific domains, the potential of diffusion models within fusion research, specifically as an alternative to PIC simulations, remains largely underexplored. Our work aims to bridge this gap by proposing Diff-PIC, a conditional diffusion model that generates high-fidelity synthetic data for fusion research. Diff-PIC offers a computationally efficient alternative to traditional PIC methods, paving the way for more scalable and resource-efficient scientific simulations.

## 6 CONCLUSIONS

This paper presents Diff-PIC, a pioneering approach that leverages the capabilities of diffusion models to generate high-fidelity synthetic data for LPI, thereby offering a computationally efficient alternative to conventional PIC simulations in nuclear fusion research. By integrating a Physically-Informed Parameter Encoder and applying the Rectified Flow Acceleration, Diff-PIC significantly augments the diffusion model's capacity to manage physical parameters, thereby accelerating the generation of high-fidelity synthetic data. These advancements facilitate rapid, resource-efficient exploration of the design space, markedly reducing the computational overhead associated with PIC simulations. Our research not only catalyzes accelerated scientific discoveries within fusion research but also sets a novel precedent for the application of generative AI models in scientific simulations. Future investigations may focus on further optimizing the distillation paradigm, harmonizing the simulation time $t_{as}$ with the diffusion time $t$, and refining the condition encoder to incorporate a wider range of physical parameters.

## ACKNOWLEDGMENTS

This work is supported by NSF under Award SHF-2326494, DMS-2015577, and DMS-2415226.

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

# A APPENDIX

## A.1 IMPLEMENTATION DETAILS OF DIFF-PIC

**Physically-informed parameter encoder (PIPE).** The PIPE module comprises two encoding strategies (i.e. positional encoders and polynomial encoders) followed by a single-layer MLP. The positional encoder (Vaswani et al., 2017), widely used in Large Language Models, utilizes sinusoidal functions to encode the input parameters, facilitating smooth interpolation. For an input parameter $x$, the positional encoder of PIPE generates a 16-dimensional embedding using sine and cosine functions at varying frequencies:

$$Pos_i(x) = \begin{cases} \sin\left(\frac{x}{10000^{\frac{2i}{d}}}\right) & \text{if } i \text{ is even,} \\ \cos\left(\frac{x}{10000^{\frac{2i}{d}}}\right) & \text{if } i \text{ is odd,} \end{cases} \tag{6}$$

where $0 \le i < 16$ and $d = 16$. Polynomial encoders include polynomial terms up to the fourth degree: $(x, x^2, x^3, x^4)$. These polynomial terms are widely used to approximate a wide range of functions effectively, capturing nonlinear correlations and unbounded growth patterns in the data, which is crucial for extrapolation. For parameters $T_e, T_i, I$, each of them is processed separately through parallel positional and polynomial encoders, concatenated as a 20-dimensional embedding, and transformed by a MLP, resulting in a refined 20-dimensional embedding. The parameter $t_{as}$ undergoes a positional encoder followed by MLP transformation, resulting in a refined 16-dimensional embedding. Finally, embeddings from these four parameters are concatenated to produce a final encoded representation, which is then combined with the simulation data as inputs to the Diff-PIC.

**The architecture and configurations of the modified U-Net backbone.** The visualization of the modified U-Net backbone is shown in Fig. 6, built using the HuggingFace Diffusers framework (von Platen et al., 2022). The U-Net backbone processes input images through an encoder-decoder architecture with skip connections. The encoding path comprises three downsampling blocks: a standard DownBlock2D module followed by two AttnDownBlock2D modules, progressively reducing spatial dimensions by $4\times$ (from H×W to H/4×W/4) while increasing the channel dimension from 64 to 256. Attention mechanisms are strategically integrated into both down-sampling and up-sampling paths to capture long-range dependencies, with the attention operations performed on flattened spatial dimensions. The decoder path mirrors the encoder with three upsampling blocks (two AttnUpBlock2D modules and one UpBlock2D module), using skip connections to preserve fine spatial details. Finally, the model applies a convolution layer to map the representation back to the original input dimension and resolution.

## A.2 DETAILS OF BASELINES

**GAN-PIC** is based on StyleGAN2 (Karras et al., 2020), a SOTA variant of Generative Adversarial Networks (GANs). It comprises two main components: a Generator and a Discriminator. The Generator in StyleGAN2 is enhanced with a style-based architecture that uses a mapping network to convert latent vectors into style vectors. These style vectors are then applied at various layers of the generator. The Discriminator's role is to differentiate between real and generated images, driving the adversarial training process that improves the fidelity of the generated outputs. **NF-PIC** is implemented based on (Zhang & Chen, 2021). NF-PIC has a drift network and a score network, both modeled using U-Net architectures. Apart from the integration of PIPE, both baselines are implemented following the experimental setups detailed in their respective original papers.

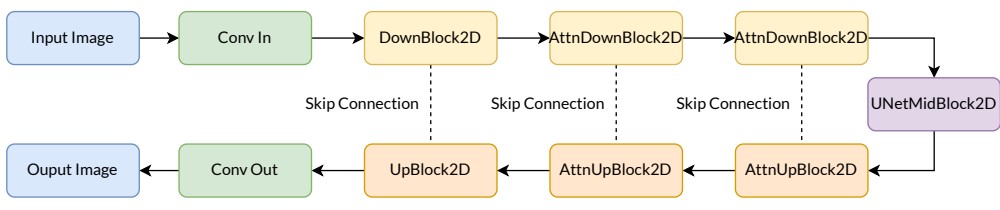

Figure 6: The U-Net backbone.

Table 4: Comparisons between MLP, Trans, and PIPE in interpolation tasks.

| Method | Training Set for E1 | | | Testing Set for E1 | | | Training Set for E2 | | | Testing Set for E2 | | |
|---|---|---|---|---|---|---|---|---|---|---|---|---|
| | MAE↓ | RMSE↓ | FID↓ | MAE↓ | RMSE↓ | FID↓ | MAE↓ | RMSE↓ | FID↓ | MAE↓ | RMSE↓ | FID↓ |
| **MLP** | 3.72e-2 | 4.16e-2 | 2.06 | 3.90e-2 | 4.57e-2 | 2.15 | 1.71e-2 | 1.92e-2 | 0.926 | 1.85e-2 | 1.98e-2 | 0.932 |
| **Trans** | 1.61e-2 | 2.80e-2 | 1.35 | 1.72e-2 | 2.31e-2 | 1.70 | 8.61e-3 | 1.02e-2 | 0.375 | 9.41e-3 | 1.18e-2 | 0.409 |
| **PIPE** | 1.56e-2 | 2.67e-2 | 1.21 | 1.68e-2 | 2.29e-2 | 1.62 | 7.95e-3 | 9.32e-3 | 0.328 | 8.26e-3 | 1.03e-2 | 0.341 |

Table 5: Comparisons between MLP, Trans, and PIPE in extrapolation tasks.

| Method | E1 10% | | | E1 20% | | | E2 10% | | | E2 20% | | |
|---|---|---|---|---|---|---|---|---|---|---|---|---|
| | MAE↓ | RMSE↓ | FID↓ | MAE↓ | RMSE↓ | FID↓ | MAE↓ | RMSE↓ | FID↓ | MAE↓ | RMSE↓ | FID↓ |
| **MLP** | 5.12e-2 | 6.01e-2 | 1.93 | 5.42e-2 | 5.64e-2 | 2.05 | 1.95e-2 | 2.62e-2 | 1.09 | 2.32e-2 | 2.91e-2 | 1.16 |
| **Trans** | 1.95e-2 | 2.61e-2 | 1.82 | 2.29e-2 | 2.73e-2 | 1.89 | 1.02e-2 | 1.55e-2 | 0.615 | 1.32e-2 | 2.04e-2 | 0.761 |
| **PIPE** | 1.83e-2 | 2.40e-2 | 1.74 | 2.18e-2 | 2.62e-2 | 1.82 | 9.47e-3 | 1.36e-2 | 0.536 | 1.13e-2 | 1.85e-2 | 0.673 |

## A.3 ABLATION STUDIES ON THE PROPOSED PIPE

To further demonstrate the effectiveness of PIPE, we conduct ablation studies by replacing the PIPE component in Diff-PIC with other commonly used encoders: MLP (denoted as MLP) and Transformer decoder (denoted as Trans). Specifically, the two-layer MLP has 128 hidden units and ReLU activations. For the Transformer decoder, we first apply a positional encoder to generate a 16-dimensional embedding for each parameter (the same as the positional encoder in PIPE). This embedding is then processed by the Transformer decoder, which comprises multi-head self-attention and a two-layer MLP with ReLU activations. The comparison results are presented in Table 4 for interpolation tasks and Table 5 for extrapolation tasks. The results show that PIPE significantly outperforms both MLP and Transformer decoder in both interpolation and extrapolation tasks. MLP struggles with both interpolation and extrapolation. The Transformer decoder, while powerful for sequence modeling and capturing relationships in discrete token sequences, is not inherently designed to effectively process continuous physical parameters, resulting in inferior performance.

To provide more insights into the effectiveness of PIPE, we conduct another ablation study to evaluate the contributions of the positional encoder (only pos), polynomial encoder (only poly) separately. The results are shown in Table 6 (interpolation tasks) and Table 7 (extrapolation tasks), highlighting the individual importance of these encoders. Specifically, combining both positional and polynomial encoders consistently outperforms using either encoder alone in both interpolation and extrapolation tasks. In interpolation tasks, the positional encoder shows better performance than the polynomial encoder, suggesting its important role in interpolation scenarios. Conversely, in extrapolation tasks, particularly at larger ranges (20%), the polynomial encoder shows relatively better performance, indicating its importance for extrapolation capabilities. These results validate our design choice of combining both encoders in PIPE to leverage their complementary strengths.

Table 6: Ablation studies on different components of PIPE in interpolation tasks.

| Method | Training Set for E1 | | | Testing Set for E1 | | | Training Set for E2 | | | Testing Set for E2 | | |
|---|---|---|---|---|---|---|---|---|---|---|---|---|
| | MAE↓ | RMSE↓ | FID↓ | MAE↓ | RMSE↓ | FID↓ | MAE↓ | RMSE↓ | FID↓ | MAE↓ | RMSE↓ | FID↓ |
| **only pos** | 1.64e-2 | 2.82e-2 | 1.37 | 1.76e-2 | 2.35e-2 | 1.72 | 8.63e-3 | 1.05e-2 | 0.379 | 9.42e-3 | 1.20e-2 | 0.412 |
| **only poly** | 1.76e-2 | 2.94e-2 | 1.52 | 1.92e-2 | 2.61e-2 | 1.85 | 1.14e-2 | 1.27e-2 | 0.517 | 1.13e-2 | 1.35e-2 | 0.546 |
| **PIPE** | 1.56e-2 | 2.67e-2 | 1.21 | 1.68e-2 | 2.29e-2 | 1.62 | 7.95e-3 | 9.33e-3 | 0.328 | 8.26e-3 | 1.03e-2 | 0.341 |

Table 7: Ablation studies on different components of PIPE in extrapolation tasks.

| Method | E1 10% | | | E1 20% | | | E2 10% | | | E2 20% | | |
|---|---|---|---|---|---|---|---|---|---|---|---|---|
| | MAE↓ | RMSE↓ | FID↓ | MAE↓ | RMSE↓ | FID↓ | MAE↓ | RMSE↓ | FID↓ | MAE↓ | RMSE↓ | FID↓ |
| **only pos** | 1.97e-2 | 2.64e-2 | 1.83 | 2.31e-2 | 2.76e-2 | 1.93 | 1.03e-2 | 1.57e-2 | 0.624 | 1.35e-2 | 2.10e-2 | 0.812 |
| **only poly** | 1.91e-2 | 2.52e-2 | 1.78 | 2.25e-2 | 2.70e-2 | 1.89 | 9.75e-3 | 1.42e-2 | 0.581 | 1.26e-2 | 1.92e-2 | 0.724 |
| **PIPE** | 1.83e-2 | 2.40e-2 | 1.74 | 2.18e-2 | 2.62e-2 | 1.82 | 9.47e-3 | 1.36e-2 | 0.536 | 1.13e-2 | 1.85e-2 | 0.673 |

