# OpenReview forum: "Diff-PIC: Revolutionizing Particle-In-Cell Nuclear Fusion Simulation with Diffusion Models"
_ICLR.cc/2025/Conference — ICLR 2025 Poster_

### Official Review · Reviewer_gLQJ · 2024-11-02

**Soundness:** 3
**Presentation:** 4
**Contribution:** 3
**Rating:** 8
**Confidence:** 4

**Summary:**

This paper introduces an approach for accelerating particle in cell simulations for laser in plasma simulations by applying a diffusion model to the computation of electric fields. These fields are produced based on key parameters in the simulation and show good accuracy when compared to literature and non-accelerated simulations.

**Strengths:**

I find the paper to be very well-written and clear. Further, the topic being approached is very important and introduced in such a way. The experiments performed to validate their method and present prior work to highlight the difference in their approach.

**Weaknesses:**

The paper is undoubtedly more geared towards nuclear physicists than for people in the ML community. What was missing for me was the advance in the ML direction. What can be taken from this work and applied to improve knowledge in the ML community? By making this connection, I feel it will be better received at ICLR.

**Questions:**

* Can you elaborate on the time encoding? At one point in the paper, it was written that the method works from snapshots. Thus, long-time series are not necessary. However, later in the paper, the authors mention that they use encoding to capture information about time. Is there some dynamic information necessary for the model function? If so, why not use a more recurrent kind of architecture? I can imagine, if dynamics are required for a complete description, that a single snapshot can occur many times in different trajectories. Therefore, a time-only encoding would not capture this information, and a state-based memory would be required. I am not an expert in these kinds of simulations, so I am likely misunderstanding something but more clarity there would be great.

---

> ### Author Response · Authors · 2024-11-22
> **Response to Reviewer gLQJ (Part 1)**
>
> We sincerely appreciate your positive feedback and your constructive suggestions. In the following, we will address your questions one by one.
>
> **The advance in the ML direction.**
>
> Thanks for your valuable comment. The novelty of our work lies in the innovative adaptation of diffusion models to address the specific challenges inherent in nuclear fusion research, extending beyond merely applying existing algorithms to a new domain. Specifically, directly adapting standard diffusion models to our problem domain does not adequately address the unique requirements of handling continuous physical conditions, which are different from common discrete and category conditions in CV. To demonstrate, we have conducted ablation studies comparing our proposed PIPE with other commonly used encoders: MLP and Transformer decoder. The results, presented in the Appendix (Table 4 and Table 5 in A.3) of the revised manuscript, demonstrate the superior performance of the introduced enhancement. For your reference, the MAE comparisons are shown in the following table, more comparison results (RMSE and FID) and discussions are presented in the Appendix (A.3).
>
> More importantly, we believe that the advancement of machine learning extends beyond algorithmic improvements, applying to real-world applications and drives progress across diverse research fields is also very important. A prime example is AlphaFold, which has revolutionized structural biology by accurately predicting protein structures, thereby accelerating biomedical research and drug discovery. Similarly, Diff-PIC exemplifies how diffusion models can benefit scientific simulations, providing researchers with powerful tools to explore and understand complex physical systems more efficiently.
> Many scientific fields require efficient simulation for exploring experimental designs and modeling. Diff-PIC mitigates the computational bottleneck associated with traditional PIC simulations, enabling more extensive and rapid exploration, accelerating progress.
>
> Besides, we release a valuable dataset of PIC simulations, facilitating further advancement in this area.
>
> **Table:**  MAE comparison between MLP, Trans, and PIPE in interpolation tasks. The lower, the better.
> | Method | E1 Training Set | E1 Testing Set | E2 Training Set | E2 Testing Set |
> |------------|------------------------------|-----------------------------|------------------------------|-----------------------------|
> | MLP   | 3.72e-2                      | 3.90e-2                     | 1.71e-2                      | 1.85e-2                     |
> | Trans  | 1.61e-2                      | 1.72e-2                     | 8.61e-3                      | 9.41e-3                     |
> | PIPE   | 1.56e-2                      | 1.68e-2                     | 7.95e-3                      | 8.26e-3                     |
>
> **Table:**  MAE comparison between MLP, Trans, and PIPE in extrapolation tasks. The lower, the better.
> | Method | E1 10% | E1 20% | E2 10% | E2 20% |
> |------------|------------------|------------------|------------------|------------------|
> | MLP    | 5.12e-2          | 5.42e-2          | 1.95e-2          | 2.32e-2          |
> | Trans  | 1.95e-2          | 2.29e-2          | 1.02e-2          | 1.32e-2          |
> | PIPE   | 1.83e-2          | 2.18e-2          | 9.47e-3           | 1.13e-2          |
>
> **What can be taken from Diff-PIC and applied to improve knowledge in the ML community?**
>
> Thanks for your insightful comment. Diff-PIC offers significant insights for other ML in science:
> 1. Using diffusion models to address common challenges in scientific computing. Our work demonstrates the power and effectiveness of the Diff-PIC for accelerating traditional, computationally expensive PIC simulations. By successfully applying our model to a challenging simulation (ignition-scale LPI simulation is a relatively complex one), we open new avenues for diffusion models in accelerating (even replacing) other scientific simulations.
>
> 2. Diff-PIC provides new insights for science problem exploration and modeling. When theoretical modeling is impractical, machine learning models like Diff-PIC can learn from data and encode the underlying physical equations in the form of model parameters.
>
> 3. We release a valuable dataset of PIC simulations, facilitating further advancement in this area.

---

> > ### Author Response · Authors · 2024-11-22
> > **Response to Reviewer gLQJ (Part 2)**
> >
> > **Elaboration on the time encoding.**
> >
> > We apologize for any confusion. In our simulations, each snapshot at a given simulation time and set of physical parameters represents a unique state of the system, and it won’t be in multiple trajectories. Specifically, each snapshot is determined by the simulation time $t_{as}$ and the other three physical parameters, these four are considered as conditions of Diff-PIC. The time encoding provides the model with information about the specific simulation time $t_{as}$ for which the snapshot is being generated. By encoding simulation time $t_{as}$ as a kind of conditions of the Diff-PIC, the model enables to generate accurate snapshots at any desired time point under a specified set of physical parameters. We choose this paradigm rather than a more recurrent kind of architecture for several reasons:
> > 1. Systemic efficiency. Diff-PIC can directly produce data for any target snapshot. This non-sequential behavior allows for substantially more efficient data generation and analysis, enabling researchers to focus on specific times of interest, breaking from the time-stepping paradigm. Even without employing a recurrent architecture, Diff-PIC effectively learns the underlying temporal dynamics of the electric fields, as evidenced by the low MAE, RMSE, and FID.
> > 2. Avoidance of Error Accumulation. Recurrent architectures may lead to the accumulation of errors over time steps—a phenomenon known as cascading errors. By treating each snapshot independently and conditioning on the specific simulation time, we avoid this issue. For Diff-PIC, each generated snapshot is based solely on the current simulation time and physical parameters, ensuring that errors do not propagate through a sequence of time steps.

---

> > > ### Comment · Reviewer_gLQJ · 2024-11-22
> > > **Response to authors**
> > >
> > > Is the argument here that a state in the system depends on an exact time? This is likely my lack of expertise in these specific kinds of simulations, however, in a typical simulation, one state of the system depends, maybe, on time and an initial condition, but not just on time itself. What role is time playing in this model? Would you say that if the three physical parameters occured at T=1 the corresponding system would appear differently than at T=10 with the same three parameters but no mention of initial conditions and an otherwise uncorrelated state?
> > >
> > > Further, if the ML model is generating different results with a change in only time but no other physical parameters, it has learned some effect of time in the system that appears nontrivial.

---

> > > > ### Author Response · Authors · 2024-11-22
> > > >
> > > > We sincerely thank the reviewer for the insightful comments and provide the following clarification.
> > > >
> > > > Our model, like traditional PIC simulations, aims to understand how the system’s data distribution evolves during the nuclear fusion process. This evolution is governed by: (1) **Time**—the duration of the fusion process after it is triggered; (2) **The intensity of the laser** that triggers the fusion process; (3) **Other physical parameters** that implicitly determine the system’s initial conditions at its equilibrium state before the fusion process begins. All of these factors are included in Diff-PIC, consistent with standard PIC simulation practices. A detailed explanation is provided below.
> > > >
> > > > The initial conditions—i.e., the equilibrium state of the system before the fusion reaction begins—are implicitly determined by physical parameters, e.g., Electron Temperature and Ion Temperature. These parameters determine the system's initial stable state prior to ignition/fusion, providing the model with the necessary information to infer the starting conditions.
> > > >
> > > > The primary focus of the PIC simulation is to study the evolution of the fusion reaction and how it impacts the system's data distribution after the laser initiates the nuclear fusion process. Laser intensity primarily drives this dynamic evolution.
> > > >
> > > > In our model, time plays a critical role in capturing the system's dynamic evolution as it transitions from a stable equilibrium state before ignition to a high-energy, dynamically evolving state during the fusion process. Diff-PIC’s ability to generate different results for the same physical parameters at different time demonstrates its capacity to accurately learn the time-driven dynamics inherent in the fusion process. This behavior aligns with the expected dynamics of nuclear fusion PIC simulations, where time serves as a key variable describing the physical evolution of the system.
> > > >
> > > > As the reviewer pointed out, this learned temporal effect is not only nontrivial but also essential for faithfully modeling the dynamics of nuclear fusion.
> > > >
> > > > We hope our answer addresses your questions. We are more than happy to provide any additional information.

---

> > > > > ### Comment · Reviewer_gLQJ · 2024-11-28
> > > > > **Response to authors**
> > > > >
> > > > > I appreciate your taking the time to address my concerns. I think, coming from a slightly different background, the way the authors introduce their "single-frame" and independence statements seemed counterintuitive as I would interpret this as being ergodic and, therefore, uncorrelated with initial conditions. However, I do not think the authors have said anything wrong, just unclear from a different perspective. In any case, my score is high, and I feel it should remain that way.

---

> > > > > > ### Author Response · Authors · 2024-11-28
> > > > > >
> > > > > > Dear Reviewer gLQJ,
> > > > > >
> > > > > > Thank you very much for your valuable feedback, and we truly appreciate the time and effort you invested in reviewing our work. Your constructive insights have significantly enhanced our manuscript, and we are delighted to receive your approval of our work. Thank you once again for your thorough and valuable review.
> > > > > >
> > > > > > Best regards,
> > > > > >
> > > > > > The Authors

---

### Official Review · Reviewer_qGUg · 2024-11-03

**Soundness:** 3
**Presentation:** 2
**Contribution:** 2
**Rating:** 5
**Confidence:** 4

**Summary:**

This paper investigates laser plasma interactions in Particle In Cell simulations (PIC) through diffusion models. They study the evolution of charged particle trajectories, influenced by the electric and magnetic fields (equation 2). The neural network designed uses the diffusion model methodology, with some novel fittings - a physically designed parameter encoder, positional encodings and legendre polynomials, and a distillation framework. The network - from what I see - takes as input electron and ion temperatures $T_e$, $T_i$ and laser intensity $I$ (i.e. $\theta = (T_e, T_i, I)$ to generate the electric field $E$, evolving it over time $t_{as}$ to obtain $E(\theta, t_{as})$. The network optimization takes place through the rectified flow acceleration technique proposed in [1, 2, 3] in equation (5).

Results are presented to show time evolution snapshots of the electric field, comparing them with two other generative models (GAN and normalizing flow). Metrics checked are MAE, FID and RMSE. They also examine the 'energy' evolution over time. They call this the interpolation and extrapolation capabilities of the generative model. Data distribution is also compared with the 'ground truth' which is obtained from a PIC simulator. The claims are that the model beats the competitors (GAN [4] and normalizing flow [5] ), and is much faster owing to the more efficient rectified flow acceleration technique used.

Post rebuttal
=========
The authors have taken great pains to explain the method and provide additional ablations and experiments, all of which supports their case that the method works rather well for PIC. The paper paper adds to the body of research work in PIC simulations:
- it works as well or better than competitive methods
- it is significantly faster than mainstream computational approaches used in plasma physics (owing to the rectified flow acceleration technique)
However, as stated before, the work does not add to novelty in terms of the method used in generative modelling and diffusion models. To this end, I am raising the score but am not (entirely) convinced about it being accepted. That being said, I think it is a marginal call.

[1] Xingchao Liu, Chengyue Gong, and Qiang Liu. Flow straight and fast: Learning to generate and
transfer data with rectified flow, 2022.

[2] Patrick Esser, Sumith Kulal, Andreas Blattmann, Rahim Entezari, Jonas Muller, Harry Saini, Yam ¨
Levi, Dominik Lorenz, Axel Sauer, Frederic Boesel, et al. Scaling rectified flow transformers for
high-resolution image synthesis. arXiv preprint arXiv:2403.03206, 2024.

[3] Xingchao Liu, Xiwen Zhang, Jianzhu Ma, Jian Peng, and Qiang Liu. Instaflow: One step is enough
for high-quality diffusion-based text-to-image generation, 2024.

[4] GAN: Alankrita Aggarwal, Mamta Mittal, and Gopi Battineni. Generative adversarial network: An overview
of theory and applications. International Journal of Information Management Data Insights, 1(1):
100004, 2021.

[5] NF: Qinsheng Zhang and Yongxin Chen. Diffusion normalizing flow. Advances in neural information
processing systems, 34:16280–16291, 2021.

**Strengths:**

- This appears to be a first of its kind (at least according to the authors) PIC simulation using generative model. I cannot verify this claim, but no other references of this type of work are compared with in the paper.

- The methodology appears to be sound, and the evaluation follows the techniques used in generative modelling generally.

- The evaluation fares quite well, compared to competitors (GAN and normalizing flow), achieving better performance and speed.

**Weaknesses:**

- Novelty: The main novelty appears to be the application of an existing method in a new domain (plasma physics). While this is not a complaint in itself, it seems (at least to me) that it is only a replacement of the modalities from more well studied setups involving images.

- Method and physical validity: It is hard for me to understand how the equation (2) (velocity field evolution) is being solved by the diffusion model. From what I see, we are learning the electric field from snapshots in time.

- Ablations: The components used in the model should, I think be ablated to show effectiveness of each piece. For instance, how does the positional embedding help? What is the performance without the Chebychev polynomial addition?

- Network Architecture: Perhaps I am missing this, but I would like to see a block containing the network design - transformer blocks, CNNs, etc.

**Questions:**

- See 'weaknesses' above: I would like clarification on whether the generative model is learning 2D snapshots of the electric field. What is the ground truth representation? Is the equation (2) at all relevant?

- The authors mention 'limited' samples. But shouldn't the PIC simulator be able to generate a large number of samples to train the model? How many samples do we think are necessary to qualify as a large enough dataset?

- Did the authors implement the alternative (GAN and NF) methods themselves? I do not see a corresponding reference for these methods for PIC (what is provided is the main method describing GAN and NF).

---

> ### Author Response · Authors · 2024-11-22
> **Response to Reviewer qGUg (Part 1)**
>
> We sincerely appreciate your insightful comments. In the following, we will address your questions one by one.
>
> **The main novelty of our work.**
>
> Thanks for your valuable comment. The novelty of our work lies in the innovative adaptation of diffusion models to address the specific challenges inherent in nuclear fusion research, extending beyond merely applying existing algorithms to a new domain. Specifically, directly adapting standard diffusion models to our problem domain does not adequately address the unique requirements of handling continuous physical conditions, which are different from common discrete and category conditions in CV. To demonstrate, we have conducted ablation studies comparing our proposed PIPE with other commonly used encoders: MLP and Transformer decoder. The results, presented in the Appendix (Table 4 and Table 5 in A.3) of the revised manuscript, demonstrate the superior performance of the introduced enhancement. For your reference, the MAE comparisons are shown in the following table, more comparison results (RMSE and FID) and discussions are presented in the Appendix (A.3).
>
> More importantly, we believe that the advancement of machine learning extends beyond algorithmic improvements, applying to real-world applications and drives progress across diverse research fields is also very important. A prime example is AlphaFold, which has revolutionized structural biology by accurately predicting protein structures, thereby accelerating biomedical research and drug discovery. Similarly, Diff-PIC exemplifies how diffusion models can benefit scientific simulations, providing researchers with powerful tools to explore and understand complex physical systems more efficiently.
> Many scientific fields require efficient simulation for exploring experimental designs and modeling. Diff-PIC mitigates the computational bottleneck associated with traditional PIC simulations, enabling more extensive and rapid exploration, accelerating progress.
>
> Besides, we release a valuable dataset of PIC simulations, facilitating further advancement in this area.
>
> **Table:**  MAE comparison between MLP, Trans, and PIPE in interpolation tasks. The lower, the better.
> | Method | E1 Training Set | E1 Testing Set | E2 Training Set | E2 Testing Set |
> |------------|------------------------------|-----------------------------|------------------------------|-----------------------------|
> | MLP   | 3.72e-2                      | 3.90e-2                     | 1.71e-2                      | 1.85e-2                     |
> | Trans  | 1.61e-2                      | 1.72e-2                     | 8.61e-3                      | 9.41e-3                     |
> | PIPE   | 1.56e-2                      | 1.68e-2                     | 7.95e-3                      | 8.26e-3                     |
>
> **Table:**  MAE comparison between MLP, Trans, and PIPE in extrapolation tasks. The lower, the better.
> | Method | E1 10% | E1 20% | E2 10% | E2 20% |
> |------------|------------------|------------------|------------------|------------------|
> | MLP    | 5.12e-2          | 5.42e-2          | 1.95e-2          | 2.32e-2          |
> | Trans  | 1.95e-2          | 2.29e-2          | 1.02e-2          | 1.32e-2          |
> | PIPE   | 1.83e-2          | 2.18e-2          | 9.47e-3           | 1.13e-2          |
>
> **Whether Diff-PIC is learning 2D snapshots of the electric fields? Is Eq. 2 relevant or being solved?**
>
> We apologize for any confusion. Yes, Diff-PIC is designed to learn and generate 2D snapshots of the electric fields involved in Laser-Plasma Interactions (LPI). While Diff-PIC does not explicitly solve the Eq. 2, it learns the data distribution generated from it, functioning as an approximation of solving the Eq. 2.
>
> **Details about the network design.**
>
> We appreciate your valuable suggestion. We have added detailed descriptions for the proposed PIPE, the architecture and configurations of the U-Net backbone, and the loss function of Diff-PIC in the Appendix (A.1) of the revised manuscript. Here, we provide a brief overview below for your reference.
>
> PIPE utilizes positional and polynomial encoding strategies followed by a MLP to generate embeddings for parameters, enabling smooth interpolation and capturing unbounded growth patterns essential for extrapolation. These embeddings are integrated with simulation data and input into a modified U-Net backbone (visualized in Fig.6). The U-Net features an encoder-decoder architecture with attention mechanisms and skip connections, comprising downsampling and upsampling blocks enhanced with attention-based ResNet blocks, GroupNorm normalization, and SiLU activations to preserve spatial details. This architecture effectively combines local convolutions with global context, making it well-suited for controlled diffusion-based generation tasks. The loss function of Diff-PIC is defined as Eq.5.

---

> > ### Author Response · Authors · 2024-11-22
> > **Response to Reviewer qGUg (Part 2)**
> >
> > **Ablation study on various pieces of the PIPE.**
> >
> > We appreciate your valuable suggestion. In the revised manuscript, we have included ablation studies on PIPE in Appendix (A.3). Specifically, we evaluate the contributions of the positional encoder (”only pos”), polynomial encoder (”only poly”)
> > separately, highlighting the individual importance of these encoders. The results show that combining both positional and polynomial encoders consistently outperforms using either encoder alone in both interpolation and extrapolation tasks. In interpolation tasks, the positional encoder demonstrates better performance than the polynomial encoder, suggesting its important role in interpolation scenarios. Conversely, in extrapolation tasks, particularly at larger ranges (20%), the polynomial encoder shows relatively better performance, indicating its importance for extrapolation capabilities. These results validate our design choice of combining both encoders in PIPE to leverage their complementary strengths. The MAE comparisons are shown as below, more comparison results (RMSE and FID) and discussions are presented in Appendix (Table 4 and Table 5 in A.3).
> >
> > **Table:** MAE comparison on different components of PIPE in interpolation tasks. The lower, the better.
> >
> > | Method | E1 Training Set | E1 Testing Set | E2 Training Set | E2 Testing Set |
> > |--------------|------------------------------|-----------------------------|------------------------------|-----------------------------|
> > | only pos | 1.64e-2                      | 1.76e-2                     | 8.63e-3                      | 9.42e-3                     |
> > | only poly| 1.76e-2                      | 1.92e-2                     | 1.14e-2                      | 1.13e-2                     |
> > | PIPE     | 1.56e-2                      | 1.68e-2                     | 7.95e-3                      | 8.26e-3                     |
> >
> > **Table:** MAE comparison on different components of PIPE in extrapolation tasks. The lower, the better.
> >
> > | Method | E1 10% | E1 20% | E2 10% | E2 20% |
> > |--------------|------------------|------------------|------------------|------------------|
> > | only pos | 1.97e-2          | 2.31e-2          | 1.03e-2          | 1.35e-2          |
> > | only poly| 1.91e-2          | 2.25e-2          | 9.75e-3           | 1.26e-2          |
> > | PIPE    | 1.83e-2          | 2.18e-2          | 9.47e-3           | 1.13e-2          |
> >
> > **Why limited samples? How many samples are qualified as a large enough dataset?**
> >
> > Thanks for your insightful question. While it is theoretically possible to generate a large number of samples using PIC simulations, in practice, there are significant computational constraints that limit this possibility. High-fidelity laser-plasma interaction (LPI) simulations are extremely computationally intensive, often requiring tens of thousands of CPU hours or more. Therefore, our goal is to train a diffusion model that can effectively learn from a limited dataset to better show the potential of Diff-PIC.
> >
> > We have compiled a dataset consisting of 6615 simulations across varied physical parameters, each simulation contains 80 snapshots of electric fields along two orthogonal directions. This amounts to over 1,000,000 samples. Given the computational constraints, the substantial size of our dataset, and the good performance achieved by Diff-PIC, we believe it qualifies as a large enough dataset in this field. We will continue generating more simulations to enhance this dataset further and make this dataset public available to foster related research.
> >
> >
> > **Implementation of the GAN-PIC and NF-PIC.**
> >
> > Thanks for your valuable suggestion. We have included implementation details for GAN-PIC, and NF-PIC in the Appendix (A.2) of the revised manuscript. Briefly, we implement GAN-PIC based on [1], and implement NF-PIC based on [2]. Both models are integrated with the proposed PIPE to enable conditional generation based on continuous physical parameters.
> >
> > [1] Tero Karras, Samuli Laine, Miika Aittala, Janne Hellsten, Jaakko Lehtinen, and Timo Aila. Analyzing and improving the image quality of stylegan. In Proceedings of the IEEE/CVF conference on computer vision and pattern recognition.
> >
> > [2] Qinsheng Zhang and Yongxin Chen. Diffusion normalizing flow. Advances in neural information processing systems.

---

> > > ### Author Response · Authors · 2024-11-24
> > > **Sincerely Seeking Advice**
> > >
> > > Dear Reviewer qGUg,
> > >
> > > We sincerely appreciate the time and effort you have dedicated to reviewing our work. Your insightful comments have provided us with valuable suggestion, and we are truly grateful for your thoughtful feedback.
> > >
> > > We understand that your schedule may be quite busy, and we would be truly grateful for the opportunity to engage in further dialogue with you during this discussion phase. We aim to ensure that our responses effectively address your concerns and to explore any additional questions or points you may have.
> > >
> > > Thank you for your thoughtful consideration.
> > >
> > > Best regards,
> > >
> > > The Authors

---

> > > > ### Comment · Reviewer_qGUg · 2024-11-26
> > > > **Thanks for comments**
> > > >
> > > > Dear authors - thanks much for the clarifications, additional experiments and thoughtful answers. The results are indeed quite convincing (though perhaps I could quibble that the pos encoding numbers are slightly marginal). I do feel that the paper would make for a new, perhaps valuable addition to the PIC literature. However, I stick to my initial view that novelty in the method used is slightly lacking. I will raise my score, but I am very much on the fence on whether we should accept.

---

> > > > > ### Author Response · Authors · 2024-11-27
> > > > >
> > > > > Dear Reviewer qGUg,
> > > > >
> > > > > We sincerely appreciate your insightful comments and valuable suggestions, which have significantly enhanced the quality of our paper. We are grateful for your recognition of our work's contribution in promoting generative AI for nuclear fusion science.
> > > > >
> > > > > Best regards,
> > > > >
> > > > > The Authors

---

### Official Review · Reviewer_4k8H · 2024-11-03

**Soundness:** 2
**Presentation:** 2
**Contribution:** 2
**Rating:** 6
**Confidence:** 2

**Summary:**

This paper proposed an new approach to tackling the computational challenges in nuclear fusion simulations by applying diffusion model. The contribution including: (1) Proposing PIPE (Physically-Informed Parameter Encoder) that improves the model's ability to understand the relationship between physical parameters and simulation outcomes. (2) Applied rectified flow acceleration. The simulated results shows this diffusion based approach significantly speed-up compared with traditional method.


========== post rebuttal

I still feel the paper's contribution on ML side is not significant enough. However, I'm less familiar with the application side. I'm lean towards to accept, if other reviewer can champion it on ML for science side.

**Strengths:**

The paper try to apply diffusion model for PARTICLE-IN-CELL NUCLEAR FUSION SIMULATION. Although I'm not sure how important of this problem, i think using diffusion model for simulation in science is an important and exciting topic. Also the the speed-up looks impressive.

**Weaknesses:**

It's unclear to me what is the ML contribution for this paper.
To me it seems most interesting part is the newly designed encoder. Can the author explain:
"Algorithmic generalization. PIPE improves the generalizability of the conditional diffusion model"
I don't quite get why such design improved generalizability compared with normal mlp/transformer layers. I hope the author would clarify it.

The application value of this paper beyond my expertise, I hope the author could also explain it and also explain how this approach generate new insights for other ML for science problem.

**Questions:**

Please see the weakness part.

Compared with previous work, "diffusion models in molecular dynamics simulations "

I hope the author can clarify:
1. The above cited paper is first paper apply diffusion to a similar domain.
2. Among the two unsolved problem, can the author explain more how this paper address "Physical soundness must be ensured"? Does it more empirical or theoriotical?
3. "Substantial efficiency improvement" I got this achieved by apply normalizing flow.

---

> ### Author Response · Authors · 2024-11-22
> **Response to Reviewer 4k8H (Part 1)**
>
> We sincerely appreciate your insightful comments. In the following, we will address your questions one by one.
>
> **The ML contribution of this paper.**
>
> Thanks for your insightful comment. The novelty of our work lies in the innovative adaptation of diffusion models to address the specific challenges inherent in nuclear fusion research, extending beyond merely applying existing algorithms to a new domain. Specifically, directly adapting standard diffusion models to our problem domain does not adequately address the unique requirements of handling continuous physical conditions, which are different from common discrete and category conditions in CV. To demonstrate, we have conducted ablation studies comparing our proposed PIPE with other commonly used encoders: MLP and Transformer decoder. The results, presented in the Appendix (Table 4 and Table 5 in A.3) of the revised manuscript, demonstrate the superior performance of the introduced enhancement. For your reference, the MAE comparisons are shown in the following table, more comparison results (RMSE and FID) and discussions are presented in the Appendix (A.3).
>
> More importantly, we believe that the advancement of machine learning extends beyond algorithmic improvements, applying to real-world applications and drives progress across diverse research fields is also very important. A prime example is AlphaFold, which has revolutionized structural biology by accurately predicting protein structures, thereby accelerating biomedical research and drug discovery. Similarly, Diff-PIC exemplifies how diffusion models can benefit scientific simulations, providing researchers with powerful tools to explore and understand complex physical systems more efficiently.
> Many scientific fields require efficient simulation for exploring experimental designs and modeling. Diff-PIC mitigates the computational bottleneck associated with traditional PIC simulations, enabling more extensive and rapid exploration, accelerating progress.
>
> Besides, we release a valuable dataset of PIC simulations, facilitating further advancement in this area.
>
> **Table:**  MAE comparison between MLP, Trans, and PIPE in interpolation tasks. The lower, the better.
> | Method | E1 Training Set | E1 Testing Set | E2 Training Set | E2 Testing Set |
> |------------|------------------------------|-----------------------------|------------------------------|-----------------------------|
> | MLP   | 3.72e-2                      | 3.90e-2                     | 1.71e-2                      | 1.85e-2                     |
> | Trans  | 1.61e-2                      | 1.72e-2                     | 8.61e-3                      | 9.41e-3                     |
> | PIPE   | 1.56e-2                      | 1.68e-2                     | 7.95e-3                      | 8.26e-3                     |
>
> **Table:**  MAE comparison between MLP, Trans, and PIPE in extrapolation tasks. The lower, the better.
> | Method | E1 10% | E1 20% | E2 10% | E2 20% |
> |------------|------------------|------------------|------------------|------------------|
> | MLP    | 5.12e-2          | 5.42e-2          | 1.95e-2          | 2.32e-2          |
> | Trans  | 1.95e-2          | 2.29e-2          | 1.02e-2          | 1.32e-2          |
> | PIPE   | 1.83e-2          | 2.18e-2          | 9.47e-3           | 1.13e-2          |
>
> **Why PIPE improves generalizability compared with normal MLP/Transformer layers?**
>
> Thanks for your insightful question. The PIPE enhances generalizability through its unique combination of positional and polynomial encoders. Positional encoders in PIPE use sinusoidal functions to map continuous input parameters into a higher-dimensional space. This method captures the relative positions of inputs in the parameter space, enabling smooth interpolation between known parameter values. Polynomial encoders in PIPE incorporate higher-order polynomial terms (e.g., $x^2$, $x^3$, $x^4$), which are widely used to approximate various kinds of complex functions. These terms allow the model to capture unbounded growth patterns, which are effective for out-of-bound extrapolation. The comparisons between PIPE, MLP, and Transformer decoder (illustrated in the response to the first question) show that, MLP struggles with both interpolation and extrapolation. The Transformer decoder, while powerful for sequence modeling and capturing relationships in discrete token sequences, is not inherently designed to effectively process continuous physical parameters, showing inferior performance.
>
> Besides, we have also included ablation studies on PIPE in Appendix (Table 4 and Table 5 in A.3), highlighting the individual importance of these encoders. The results validate our design choice of combining both encoders in PIPE to leverage their complementary strengths.

---

> > ### Author Response · Authors · 2024-11-22
> > **Response to Reviewer 4k8H (Part 2)**
> >
> > **The application value of Diff-PIC and its new insights for other ML in science.**
> >
> > Thanks for your insightful comment. Diff-PIC offers significant application value. Specifically, Diff-PIC addresses a critical bottleneck in nuclear fusion ignition research: the time-consuming and expensive simulations that hinder progress. By demonstrating that diffusion models can accelerate these simulations, we contribute to the advancement of nuclear fusion ignition research. Given that nuclear fusion is very promising for achieving a sustainable energy source (National Ignition Facility has generated 3.4 MJ and 5.2 MJ energy from 2.2 MJ input energy, respectively [1]), our work has far-reaching implications for humanity.
> >
> > Diff-PIC also shows new insights for other ML in science:
> > 1. Using diffusion models to address common challenges in scientific computing. Our work demonstrates the power and effectiveness of the Diff-PIC for accelerating traditional, computationally expensive PIC simulations. By successfully applying our model to a challenging simulation (ignition-scale LPI simulation is a relatively complex one), we open new avenues for diffusion models in accelerating (even replacing) other scientific simulations.
> >
> > 2. Diff-PIC provides new insights for science problem exploration and modeling. When theoretical modeling is impractical, machine learning models like Diff-PIC can learn from data and encode the underlying physical equations in the form of model parameters.
> >
> >
> > **Clarification on the cited paper “diffusion models in molecular dynamics simulations”.**
> >
> > We apologize for any confusion. We cite them to refer recent studies that apply diffusion models to molecular dynamics simulations to highlight the broader applicability of diffusion models in scientific data generation, and meanwhile, positioning our work within the growing body of research that leverages diffusion models for scientific simulations.
> >
> > **How Diff-PIC ensures physical soundness?**
> >
> > Thanks for your insightful question. Current state-of-the-art modeling method suggests that an approximately 12% prediction error is effective [2]. In our experiments, Diff-PIC achieves MAE and RMSE values of approximately 1-2% relative to the ground truth data. Such precision is sufficient for the generated data to be useful in nuclear fusion research, such as preliminary analyses, parameter exploration, and prediction modeling.
> >
> > **Clarification on substantial efficiency improvement.**
> >
> > We apologize for any confusion. The substantial efficiency improvement of Diff-PIC is achieved by comparing with traditional PIC simulations for a 100 picosecond simulation. While GAN-PIC and NF-PIC also offer computational benefits over traditional simulations, their accuracy is significantly lower than that of Diff-PIC.
> >
> >
> > [1] https://lasers.llnl.gov/science/achieving-fusion-ignition
> >
> > [2] Ejaz, R., Gopalaswamy, V., Lees, A., Kanan, C., Cao, D., & Betti, R. (2024). Deep learning-based predictive models for laser direct drive at the Omega Laser Facility. Physics of Plasmas, 31(5).

---

> > > ### Author Response · Authors · 2024-11-24
> > > **Sincerely Seeking Advice**
> > >
> > > Dear Reviewer 4k8H,
> > >
> > > We sincerely appreciate the time and effort you have dedicated to reviewing our work. Your insightful comments have provided us with valuable suggestion, and we are truly grateful for your thoughtful feedback.
> > >
> > > We understand that your schedule may be quite busy, and we would be truly grateful for the opportunity to engage in further dialogue with you during this discussion phase. We aim to ensure that our responses effectively address your concerns and to explore any additional questions or points you may have.
> > >
> > > Thank you for your thoughtful consideration.
> > >
> > > Best regards,
> > >
> > > The Authors

---

### Official Review · Reviewer_Mxd3 · 2024-11-04

**Soundness:** 2
**Presentation:** 3
**Contribution:** 2
**Rating:** 6
**Confidence:** 2

**Summary:**

Paper uses a diffusion model to generate data simulating a Particle-in-cell simulation for nuclear fusion research. The diffusion model takes parameters (electron temperature, ion temperature, and laser intensity) and timestep and generates an electric field snapshot. The paper designs a parameter encoder which uses positional encoding for timestep, and positional encoding concatenated with a polynomial transformation for electron temperature, ion temperature, and laser intensity. The model is trained with rectified flow-based acceleration using a u-net for learned trajectory. Paper includes experiments of quantitative interpolation, extrapolation, energy evaluation vs snapshot, qualitative comparison with ground truth data, and speedup vs particle-in-cell simulation.

**Strengths:**

The paper is original in using diffusion model for simulating particle-in-cell data. Paper tackles an important problem of design in nuclear fusion research. Paper is clearly written.

**Weaknesses:**

A weakness is that it is unclear how close the generated data should be to the true data to be useful for nuclear fusion design/research. hard to evaluate the scale of quantitative error. also do the fine details matter in the E field? because they do not look close in Fig 4 visualization. could authors provide context on what level of accuracy is required for the generated data to be useful in nuclear fusion research, and discuss the importance of fine details in the electric field and how this impacts the utility of their approach.

the main conclusion from paper's experiments seems to be that diffusion with u-net performs better than Gan or normalizing flow, for this task.

**Questions:**

Perhaps authors could add an ablation study on various pieces of the physically informed parameter encoder? such as the positional encoding or polynomial transformation

Perhaps authors could add uncertainty estimates to numbers in Tables? such as standard deviation over multiple runs or confidence intervals

Can authors add energy evaluation on Gan-Pic and NF-Pic onto Fig 5? Would be good to have same energy evaluation metrics (MAE and RMSE) included for GAN-PIC and NF-PIC in Fig 5, allowing for a direct comparison across all methods

---

> ### Author Response · Authors · 2024-11-22
> **Response to Reviewer Mxd3 (Part 1)**
>
> We sincerely appreciate your insightful comments. In the following, we will address your questions one by one.
>
> **What level of accuracy is required for the generated data to be useful in nuclear fusion research?**
>
> Thanks for your insightful question. Current state-of-the-art modeling method suggests an approximately 12% prediction error is effective [1]. In our experiments, Diff-PIC achieves MAE and RMSE values of approximately 1-2% relative to the ground truth data. Such precision is sufficient for the generated data to be useful in nuclear fusion research, such as preliminary analyses, parameter exploration, and prediction modeling.
>
> **The scale of the quantitative error.**
>
> Thanks for your valuable comment. We have normalized the dataset to [0,1] range in our experiments to make the reported MAE, RMSE, and FID metrics directly indicate the scale of the quantitative errors (lines 297-299).
>
> **The importance of fine details in the electric fields and how it impacts the utility of Diff-PIC.**
>
> Thanks for your insightful question. The fine details in the electric fields do not affect the utility of Diff-PIC. For a given set of conditions, PIC simulations will produce different outcomes from different runs, depending on the "random seed" used in the input deck (there are randomness in PIC simulations). This means that our task is actually a distribution learning problem rather than a point estimation. Diffusion models are well-suited for this task as they are designed to effectively learn data distributions.
>
> **The main conclusion of our work.**
>
> Thanks for your insightful comment. The novelty of our work lies in the innovative adaptation of diffusion models to address the specific challenges inherent in nuclear fusion research, extending beyond merely applying existing algorithms to a new domain. Specifically, directly adapting standard diffusion models to our problem domain does not adequately address the unique requirements of handling continuous physical conditions, which are different from common discrete and category conditions in CV. To demonstrate, we have conducted ablation studies comparing our proposed PIPE with other commonly used encoders: MLP and Transformer decoder. The results, presented in the Appendix (Table 4 and Table 5 in A.3) of the revised manuscript, demonstrate the superior performance of the introduced enhancement. For your reference, the MAE comparisons are shown in the following table, more comparison results (RMSE and FID) and discussions are presented in the Appendix (A.3).
>
> More importantly, we believe that the advancement of machine learning extends beyond algorithmic improvements, applying to real-world applications and drives progress across diverse research fields is also very important. A prime example is AlphaFold, which has revolutionized structural biology by accurately predicting protein structures, thereby accelerating biomedical research and drug discovery. Similarly, Diff-PIC exemplifies how diffusion models can benefit scientific simulations, providing researchers with powerful tools to explore and understand complex physical systems more efficiently.
> Many scientific fields require efficient simulation for exploring experimental designs and modeling. Diff-PIC mitigates the computational bottleneck associated with traditional PIC simulations, enabling more extensive and rapid exploration, accelerating progress.
>
> Besides, we release a valuable dataset of PIC simulations, facilitating further advancement in this area.
>
> **Table:**  MAE comparison between MLP, Trans, and PIPE in interpolation tasks. The lower, the better.
> | Method | E1 Training Set | E1 Testing Set | E2 Training Set | E2 Testing Set |
> |------------|------------------------------|-----------------------------|------------------------------|-----------------------------|
> | MLP   | 3.72e-2                      | 3.90e-2                     | 1.71e-2                      | 1.85e-2                     |
> | Trans  | 1.61e-2                      | 1.72e-2                     | 8.61e-3                      | 9.41e-3                     |
> | PIPE   | 1.56e-2                      | 1.68e-2                     | 7.95e-3                      | 8.26e-3                     |
>
> **Table:**  MAE comparison between MLP, Trans, and PIPE in extrapolation tasks. The lower, the better.
> | Method | E1 10% | E1 20% | E2 10% | E2 20% |
> |------------|------------------|------------------|------------------|------------------|
> | MLP    | 5.12e-2          | 5.42e-2          | 1.95e-2          | 2.32e-2          |
> | Trans  | 1.95e-2          | 2.29e-2          | 1.02e-2          | 1.32e-2          |
> | PIPE   | 1.83e-2          | 2.18e-2          | 9.47e-3           | 1.13e-2          |
>
> [1] Ejaz, R., Gopalaswamy, V., Lees, A., Kanan, C., Cao, D., & Betti, R. (2024). Deep learning-based predictive models for laser direct drive at the Omega Laser Facility. Physics of Plasmas, 31(5).

---

> > ### Author Response · Authors · 2024-11-22
> > **Response to Reviewer Mxd3 (Part 2)**
> >
> > **Ablation study on various pieces of the PIPE.**
> >
> > We appreciate your valuable suggestion. In the revised manuscript, we have included ablation studies on PIPE in Appendix (A.3). Specifically, we evaluate the contributions of the positional encoder (”only pos”), polynomial encoder (”only poly”)
> > separately, highlighting the individual importance of these encoders. The results show that combining both positional and polynomial encoders consistently outperforms using either encoder alone in both interpolation and extrapolation tasks. In interpolation tasks, the positional encoder demonstrates better performance than the polynomial encoder, suggesting its important role in interpolation scenarios. Conversely, in extrapolation tasks, particularly at larger ranges (20%), the polynomial encoder shows relatively better performance, indicating its importance for extrapolation capabilities. These results validate our design choice of combining both encoders in PIPE to leverage their complementary strengths. The MAE comparisons are shown as below, more comparison results (RMSE and FID) and discussions are presented in Appendix (Table 4 and Table 5 in A.3).
> >
> > **Table:** MAE comparison on different components of PIPE in interpolation tasks. The lower, the better.
> >
> > | Method | E1 Training Set | E1 Testing Set | E2 Training Set | E2 Testing Set |
> > |--------------|------------------------------|-----------------------------|------------------------------|-----------------------------|
> > | only pos | 1.64e-2                      | 1.76e-2                     | 8.63e-3                      | 9.42e-3                     |
> > | only poly| 1.76e-2                      | 1.92e-2                     | 1.14e-2                      | 1.13e-2                     |
> > | PIPE     | 1.56e-2                      | 1.68e-2                     | 7.95e-3                      | 8.26e-3                     |
> >
> > **Table:** MAE comparison on different components of PIPE in extrapolation tasks. The lower, the better.
> >
> > | Method | E1 10% | E1 20% | E2 10% | E2 20% |
> > |--------------|------------------|------------------|------------------|------------------|
> > | only pos | 1.97e-2          | 2.31e-2          | 1.03e-2          | 1.35e-2          |
> > | only poly| 1.91e-2          | 2.25e-2          | 9.75e-3           | 1.26e-2          |
> > | PIPE    | 1.83e-2          | 2.18e-2          | 9.47e-3           | 1.13e-2          |
> >
> >
> > **Add uncertainty estimates to numbers in Tables.**
> >
> > We appreciate your valuable suggestion. In response, we are currently running more experiments using different random seeds to calculate the standard deviation for uncertainty estimate. Once finished, we will add revisions to the manuscript.
> >
> > **Add GAN-PIC and NF-PIC to Fig. 5.**
> >
> > Thanks for your valuable suggestion. We have updated Figure 5 in the revised manuscript for a direct comparison across all methods.

---

> > > ### Author Response · Authors · 2024-11-24
> > > **Sincerely Seeking Advice**
> > >
> > > Dear Reviewer Mxd3,
> > >
> > > We sincerely appreciate the time and effort you have dedicated to reviewing our work. Your insightful comments have provided us with valuable suggestion, and we are truly grateful for your thoughtful feedback.
> > >
> > > We understand that your schedule may be quite busy, and we would be truly grateful for the opportunity to engage in further dialogue with you during this discussion phase. We aim to ensure that our responses effectively address your concerns and to explore any additional questions or points you may have.
> > >
> > > Thank you for your thoughtful consideration.
> > >
> > > Best regards,
> > >
> > > The Authors

---

> > > > ### Comment · Reviewer_Mxd3 · 2024-11-26
> > > >
> > > > Thanks to the authors for response. I think the context of scale of error, fine details of E field, direct comparison in Fig. 5 and uncertainty estimates will improve paper, so increased my score.

---

> > > > > ### Author Response · Authors · 2024-11-26
> > > > >
> > > > > Dear Reviewer Mxd3,
> > > > >
> > > > > Thank you for your thoughtful feedback and for taking the time to review our response. We sincerely appreciate your acceptance of our work. Your constructive insights have been invaluable in improving the quality of the manuscript. Please do not hesitate to let us know if there are any remaining concerns or additional details that we can address to further improve the manuscript.
> > > > >
> > > > > Thank you once again for your thorough and valuable review.
> > > > >
> > > > > Best regards,
> > > > >
> > > > > The Authors

---

### Official Review · Reviewer_FMwc · 2024-11-04

**Soundness:** 4
**Presentation:** 3
**Contribution:** 4
**Rating:** 8
**Confidence:** 3

**Summary:**

The paper describes a computationally efficient alternative to Particle-in-Cell (PIC) simulations for nuclear fusion research, specifically for generating Laser-Plasma Interaction (LPI) data. This is achieved through the application of a diffusion model, designed to replace the resource-intensive and time-consuming nature of PIC simulations by introducing a Physically-Informed Parameter Encoder and the Rectified Flow Technique. The work primarily aims to advance nuclear fusion research by making high-fidelity data generation feasible at a fraction of the traditional computational cost.

**Strengths:**

The submission is clear in its objectives and introduces novel elements like the Physically-Informed Parameter Encoder (PIPE) and Rectified Flow Acceleration (RFA) Technique, which together can generate high-fidelity data by applying a diffusion model. The experimental setup is robust, providing a detailed comparison of Diff-PIC against state-of-the-art generative models (GANs and Normalizing Flow) on multiple performance metrics (MAE, RMSE, FID). The authors demonstrate significant speedups and error reductions, substantiating the model’s effectiveness with quantitative evidence. Diff-PIC makes a strong alternative to traditional Particle-in-Cell (PIC) simulations, since it achieves orders-of-magnitude speedup, while retaining high fidelity and reducing the computational cost of PIC.

The PIPE provides a novel way to embed physical constraints, ensuring the generated data remains physically consistent without modifying the core diffusion process. This approach keeps the model flexible, allowing it to adapt to various scenarios, such as handling different particle types (e.g., electrons or ions), while avoiding the complexity of embedding physics-informed parameters directly within the diffusion model itself. While RFA is becoming more known in specific domains like high-resolution image generation, it remains novel in scientific applications, particularly for complex simulations like PIC. The inclusion of RFA in Diff-PIC enhances efficiency, offering significant speedup without sacrificing fidelity for generating realistic. Despite some missing details on specific model components or implementation—the paper’s contributions are substantial and have the potential to drive advancements in scientific data generation for PIC simulations and could become a tool for the broader community.

**Weaknesses:**

The authors discuss how diffusion models align with the problem and highlight PIPE as a benefit for Diff-PIC, but GAN-PIC and NF-PIC include PIPE in their comparisons as well. Also, Diff-PIC’s performance heavily depends on PIPE’s ability to generalize from limited training data, which may require retraining or fine-tuning the encoder if the physical parameter ranges shift. While the paper evaluates Diff-PIC using error metrics and continuity checks, it would benefit from additional validation against physics-specific benchmarks and more interpretability studies. Also, architectural comparisons to GAN-PIC and NF-PIC in the LPI data generation context would provide insights into the superior performance of Diff-PIC.

**Questions:**

The paper introduces innovative components like the PIPE with polynomial encodings, a U-Net for score-based modeling in the RFA. However, it lacks specific details on critical elements: it does not fully describe the polynomial encoder (such as the choice of polynomial degree or basis functions), the architecture and configurations of the U-Net, or the exact loss function guiding the training process. These missing details limit a clear understanding of how each component is optimized for effective high-fidelity data generation.

---

> ### Author Response · Authors · 2024-11-22
> **Response to Reviewer FMwc (Part 1)**
>
> We sincerely appreciate your positive feedback and your constructive suggestions. In the following, we will address your questions one by one.
>
> **Why incorporate PIPE into GANs and NF?**
>
> Thanks for your insightful question. We integrate the PIPE into both the GAN and NF for two main reasons:
>
> 1. Enabling Conditional Generation. The selected GAN and NF are designed for unconditional generation and lack the ability to directly handle continuous physical parameters required for our task. Incorporating PIPE equips these models with the capacity to generate data conditioned on specific parameters.
>
> 2. Ensuring Controlled and Fair Comparison. Using PIPE across all models standardizes the parameter encoding process, eliminating biases that could arise from using different encoding techniques for each method.
>
> We agree that ablation studies are needed to demonstrate the effectiveness of PIPE, thus we conduct an ablation study comparing PIPE with other commonly used encoders: MLP and Transformer decoder (denoted as Trans). The MAE comparisons are shown as below, more comparison results (RMSE and FID) and discussions are presented in the Appendix (Table 4 and Table 5 in A.3) of the revised manuscript. The results show that PIPE significantly outperforms both MLP and Transformer decoder in both interpolation and extrapolation tasks. MLP struggles with both interpolation and extrapolation. The Transformer decoder, while powerful for sequence modeling and capturing relationships in discrete token sequences, is not inherently designed to effectively process continuous physical parameters, showing inferior performance.
>
> **Table:**  Comparisons between MLP, Trans, and PIPE in interpolation tasks. The lower, the better.
> | Method | E1 Training Set | E1 Testing Set | E2 Training Set | E2 Testing Set |
> |------------|------------------------------|-----------------------------|------------------------------|-----------------------------|
> | MLP    | 3.72e-2                      | 3.90e-2                     | 1.71e-2                      | 1.85e-2                     |
> | Trans  | 1.61e-2                      | 1.72e-2                     | 8.61e-3                      | 9.41e-3                     |
> | PIPE   | 1.56e-2                      | 1.68e-2                     | 7.95e-3                      | 8.26e-3                     |
>
> **Table:**  Comparisons between MLP, Trans, and PIPE in extrapolation tasks. The lower, the better.
> | Method | E1 10% | E1 20% | E2 10% | E2 20% |
> |------------|------------------|------------------|------------------|------------------|
> | MLP    | 5.12e-2          | 5.42e-2          | 1.95e-2          | 2.32e-2          |
> | Trans  | 1.95e-2          | 2.29e-2          | 1.02e-2          | 1.32e-2          |
> | PIPE   | 1.83e-2          | 2.18e-2          | 9.47e-3           | 1.13e-2          |
>
> **Require fine-tuning or retaining when the parameter range shifts.**
>
> Thanks for your valuable comment. Diff-PIC enables effective interpolation and extrapolation among certain parameter ranges, which is sufficient for exploring the effective parameter range in fusion research. Fine-tuning or retraining is required when adapting the proposed method to other fusion applications or other fields with different physical parameters.
>
> **Details on the components of Diff-PIC.**
>
> We appreciate your valuable suggestion. We have added detailed descriptions for the proposed PIPE, the architecture and configurations of the U-Net backbone, and the loss function of Diff-PIC in the Appendix (A.1) of the revised manuscript. We provide a brief overview below for your reference.
>
> PIPE utilizes positional and polynomial encoding strategies followed by a MLP to generate embeddings for parameters, enabling smooth interpolation and capturing unbounded growth patterns essential for extrapolation. These embeddings are integrated with simulation data and input into a modified U-Net backbone (visualized in Fig.6). The U-Net features an encoder-decoder architecture with attention mechanisms and skip connections, comprising downsampling and upsampling blocks enhanced with attention-based ResNet blocks, GroupNorm normalization, and SiLU activations to preserve spatial details. This architecture effectively combines local convolutions with global context, making it well-suited for controlled diffusion-based generation tasks. The loss function of Diff-PIC is defined as Eq.5.

---

> > ### Author Response · Authors · 2024-11-22
> > **Response to Reviewer FMwc (Part 2)**
> >
> > **Architectural comparisons between Diff-PIC and GAN-PIC and NF-PIC**
> >
> > Thanks for your valuable suggestion. We have included detailed architectural comparisons between Diff-PIC, GAN-PIC, and NF-PIC in the Appendix (A.2) of the revised manuscript. We provide a brief overview below for your reference.
> >
> > Diff-PIC employs a modified U-Net architecture (detailed in Appendix A.1). GAN-PIC is based on StyleGAN2 [1], a SOTA variant of Generative Adversarial Networks. It comprises two main components: a Generator and a Discriminator. The Generator in StyleGAN2 is enhanced with a style-based architecture that uses a mapping network to convert latent vectors into style vectors. These style vectors are then applied at various layers of the generator through adaptive instance normalization, allowing precise control over image attributes at different scales. The Discriminator’s role is to differentiate between real and generated images, driving the adversarial training process that improves the realism and fidelity of the generated outputs. Additionally, StyleGAN2 incorporates architectural innovations such as progressive growing and skip connections, which contribute to training stability and high-quality image synthesis. NF-PIC is implemented based on [2], belong to normalizing flow based generative models. NF-PIC has a drift network and a score network, both modeled using U-Net architectures. But different from diffusion models, it relies on a sequence of invertible and differentiable transformations to construct complex distributions from simple ones, which are computationally intensive due to the need for invertibility and calculating the Jacobian determinant.
> >
> >
> > [1] Tero Karras, Samuli Laine, Miika Aittala, Janne Hellsten, Jaakko Lehtinen, and Timo Aila. Analyzing and improving the image quality of stylegan. In Proceedings of the IEEE/CVF conference on computer vision and pattern recognition.
> >
> > [2] Qinsheng Zhang and Yongxin Chen. Diffusion normalizing flow. Advances in neural information processing systems.

---

> > > ### Comment · Reviewer_FMwc · 2024-12-02
> > > **Response to authors**
> > >
> > > Thank you for taking the time to address my comments and incorporating it into your revisions. I appreciate the detailed clarifications you provided on the incorporation of PIPE into GANs and NFs, and the architectural specifics of Diff-PIC.
> > >
> > > Your work offers a significant step forward in the field, providing a more efficient approach compared to traditional simulations and other methods. I’ll maintain my positive score.

---

> > > > ### Author Response · Authors · 2024-12-03
> > > >
> > > > Dear Reviewer FMwc,
> > > >
> > > > We sincerely thank you for your thoughtful feedback and for dedicating your time to review our manuscript. We greatly appreciate your approval of our work and are grateful for your constructive insights, which have significantly enhanced the quality of our work.
> > > >
> > > > Best Regards,
> > > >
> > > > The Authors

---

### Comment · Area_Chair_uF7j · 2024-11-25
**Please engage in the discussion**

Dear all,

Many thanks to the reviewers for their constructive reviews and the authors for their detailed responses. Please use the next ~2 days to discuss any remaining queries as the discussion period is about to close.

thank you.

Regards,

AC

---

### Author Response · Authors · 2024-12-03
**Summary of Rebuttal and Discussion**

Dear Area Chair and Reviewers,

We sincerely appreciate the constructive feedback and insightful suggestions throughout the review process. We are pleased to note the positive outcomes from the discussion phase:

- Reviewer FMwc upheld the high score, acknowledging that our detailed responses effectively addressed their questions and affirming the contribution and significance of Diff-PIC in scientific data generation.
- Reviewer Mxd3 increased the score following our added ablation studies on the proposed PIPE, clarifications on the contributions, and demonstrations of the practical value of our work in scientific studies. We are grateful to the reviewer for regarding our work as presenting an original idea and tackling an important problem in nuclear fusion research.
- Reviewer 4k8H retained their positive score. We appreciate the reviewer for regarding the use of diffusion models for scientific simulations as important and exciting.
- Reviewer qGUg increased the score, acknowledging the detailed clarifications, additional ablation studies, and convincing results. We thank the reviewer for considering our work a new and valuable addition to the community.
- Reviewer gLQJ maintained the high score, valuing the detailed answers that effectively address their concerns.  We are thankful to the reviewer for considering our work to be well-written and important.

As the discussion concludes, we would like to summarize the key points addressed:

1. Novelty of Diff-PIC and Its ML Contribution.

- We emphasized Diff-PIC's novelty in proposing innovative adaptations of diffusion models to address the challenges in nuclear fusion research--the computationally expensive PIC simulations that hinder progress. We conducted additional experiments to validate the proposed PIPE by comparing it with other commonly used parameter encoders, including MLP and Transformer decoders. The results, detailed in Appendix A.3, highlight the superior performance of the proposed adaptation PIPE in handling continuous physical parameters.
- Diff-PIC exemplifies how diffusion models can benefit scientific simulations, providing researchers with powerful tools to explore and understand complex physical systems more efficiently. We highlighted that a key advancement of ML is adapting advanced methods to real-world applications, driving progress across diverse research fields—-as seen with breakthroughs like AlphaFold.

2. Further Insights into the PIPE Design.
- We conducted ablation studies to evaluate different components of PIPE in contributing the interpolation and extrapolation tasks, further validating our design choice in Appendix A.3.
3. Significant Value of Diff-PIC in Nuclear Fusion Research.
- We expanded the Discussion section to provide deeper insights into the significant value of Diff-PIC in nuclear fusion research. We referenced a SOTA fusion modeling method, which suggests a prediction error of approximately 12% is considered effective. Diff-PIC achieves approximately 1-2% error, which is sufficient for preliminary analysis, parameter exploration, and prediction modeling.

4. Improved Explanation and Presentation.
- We provided detailed descriptions and implementations of Diff-PIC and the selected baselines to enhance clarity and reproducibility in Appendix A.1 and A.2.

We would like to emphasize the **contributions** of Diff-PIC as recognized by reviewers:

- **Significance:** (1) *Accelerating Scientific Progress:* Diff-PIC promotes scientific advancement by making high-fidelity simulations faster and at a fraction of the computational cost of traditional PIC simulations, thereby accelerating scientific progress. (2) *Expanding Applications of Generative AI:* Diff-PIC demonstrates the potential and effectiveness of diffusion models in scientific simulations, expanding the applications of generative AI. Diff-PIC could become a valuable tool and offer new insights for both the scientific and ML communities.

- **Novelty:** The proposed PIPE introduces a novel way to embed continuous physical parameters that are different from commonly discrete and categorical parameters in CV, while keeping the overall framework flexible and adaptable to other scenarios. The integration of rectified flow acceleration further enhances efficiency, offering significant speedup without sacrificing fidelity.

- **Dataset Contribution:** We introduced a substantial dataset consisting of over 1,000,000 samples across varied physical parameters, facilitating further research and exploration in the direction of using GenAI for Science.

Finally, we thank the reviewers for their invaluable comments, which have strengthened our work. We are confident that Diff-PIC provides meaningful insights and advances the development of diffusion models in scientific data generation.

Best Regards,

The Authors

---

### Meta-Review · Area_Chair_uF7j · 2024-12-20

**Metareview:**

The paper introduces Diff-PIC, a novel framework that leverages diffusion models to generate high-fidelity simulation data for Laser-Plasma Interaction (LPI) in nuclear fusion. Diff-PIC addresses the computational bottleneck of traditional Particle-in-Cell (PIC) simulations by offering a 16,200× speedup while maintaining high accuracy. The paper contributes two key things: 1) a Physically-Informed Parameter Encoder (PIPE), which captures relationships between physical parameters and PIC data, enabling smooth interpolation and extrapolation of unseen parameter values; and 2) a Rectified Flow Acceleration (RFA) technique that allows for one-step data generation, further enhancing efficiency. The authors also provide an open-source dataset of 6,615 LPI simulations, positioning Diff-PIC as a new baseline for scientific data generation.

In terms of the strengths of the approach, as highlighted by the reviewers as well, include very good computational efficiency, strong generalisation to new parameter ranges, and a reduction in computational energy usage by 10,000×. Diff-PIC outperforms state-of-the-art methods like GAN-PIC and NF-PIC in terms of mean absolute error (MAE), root mean squared error (RMSE), and Fréchet Inception Distance (FID), showing some strong physical validity in its generated data.

However a minor weakness concerns the framework’s generalisation which is only tested on limited setups, therefore restricting its wider impact on machine learning methodologies.

Overall, Diff-PIC represents a significant step forward for ML applications in nuclear fusion research, providing an efficient and accurate alternative to traditional PIC simulations, with the potential to accelerate research in fusion and other areas requiring computationally intensive simulations.

**Additional Comments On Reviewer Discussion:**

Overall the five reviewers have been positive towards this paper. They have acknowledged the domain-specific contributions of this paper to nuclear fusion research which have several elements, both in terms of providing machine learning methodologies and in providing improvements in simulation speedups and dataset release.

As is common with such papers, reviewers queried the technical novelty of this paper, However, it needs to be highlighted that there is a novel technical component in that they propose a physics-informed parameter encoder that captures the relationships between physical parameters and Particle-in-Cell simulations.

---

### Decision · Program_Chairs · 2025-01-22

Accept (Poster)